# Single-cell sequencing of neonatal uterus reveals an Misr2+ endometrial progenitor indispensable for fertility

Hatice Duygu Saatcioglu[1,2], Motohiro Kano[1,2], Heiko Horn[1,2,3], Lihua Zhang[1,2], Wesley Samore[4], Nicholas Nagykery[1,2], Marie-Charlotte Meinsohn[1,2], Minsuk Hyun[5], Rana Suliman[1,2], Joy Poulo[1,2,3], Jennifer Hsu[6], Caitlin Sacha[6], Dan Wang[7], Guangping Gao[7], Kasper Lage[1,2,3], Esther Oliva[4], Mary E Morris Sabatini[8], Patricia K Donahoe[1,2], David Pépin[1,2]*

[1]Pediatric Surgical Research Laboratories, Massachusetts General Hospital, Boston, United States; [2]Department of Surgery, Harvard Medical School, Boston, United States; [3]Stanley Center, Broad Institute of MIT and Harvard, Cambridge, United States; [4]Department of Pathology, Massachusetts General Hospital, Boston, United States; [5]Department of Neurobiology, Harvard Medical School, Boston, United States; [6]Department of Gynecology and Reproductive Biology, Massachussets General Hospital, Boston, United States; [7]Horae Gene Therapy Center, University of Massachusetts Medical School, Worcester, United States; [8]Department of Gynecology and Reproductive Biology, Massachussets General Hospital, Boston, United States

*For correspondence:
dpepin@mgh.harvard.edu

Competing interests: The authors declare that no competing interests exist.

**Abstract** The Mullerian ducts are the anlagen of the female reproductive tract, which regress in the male fetus in response to MIS. This process is driven by subluminal mesenchymal cells expressing Misr2, which trigger the regression of the adjacent Mullerian ductal epithelium. In females, these Misr2+ cells are retained, yet their contribution to the development of the uterus remains unknown. Here, we report that subluminal Misr2+ cells persist postnatally in the uterus of rodents, but recede by week 37 of gestation in humans. Using single-cell RNA sequencing, we demonstrate that ectopic postnatal MIS administration inhibits these cells and prevents the formation of endometrial stroma in rodents, suggesting a progenitor function. Exposure to MIS during the first six days of life, by inhibiting specification of the stroma, dysregulates paracrine signals necessary for uterine development, eventually resulting in apoptosis of the Misr2+ cells, uterine hypoplasia, and complete infertility in the adult female.
DOI: https://doi.org/10.7554/eLife.46349.001

## Introduction

In mammals, both sexes initially develop Mullerian ducts, consisting of a single layer of epithelial cells surrounded by undifferentiated mesenchymal cells. Mullerian Inhibiting Substance Receptor (Misr2/Amhr2) expression is first detected in the Mullerian mesenchyme at around E13.5 both in males and in females (*Arango et al., 2008*). In male mice, secretion of MIS (also known as Anti-Mullerian Hormone or AMH) by the developing testes causes regression of the Mullerian ducts during embryonic days 14.5–15.5 (*Jost, 1947*; *Josso et al., 1976*). Regression of the male Mullerian ductal epithelium is mediated through non-cell autonomous paracrine signals emanating from the underlying Misr2+ mesenchymal cells in response to MIS, and is thought to be dependent on Wnt signaling and beta-catenin activity in the epithelium (*Mullen and Behringer, 2014*). The Mis/Misr2 (Amh/

**eLife digest** In the womb, mammals possess all of the preliminary sexual structures necessary to become either male or female. This includes the Mullerian duct, which develops into the Fallopian tubes, uterus, cervix, and vagina in female fetuses. In male fetuses, the testis secretes a hormone called Mullerian inhibiting substance (MIS). This triggers the activity of a small group of cells, known as Misr2+ cells, that cause the Mullerian duct to degenerate, preventing males from developing female sexual organs.

It was not clear what happens to Misr2+ cells in female fetuses or if they affect how the uterus develops. Saatcioglu et al. now show that in newborn female mice and rats, a type of Misr2+ cell that sits within a thin inner layer of the developing uterus still responds to MIS. At this time, the uterus is in a critical early period of development. Treating the mice and rats with MIS protein during their first six days of life eventually caused the Misr2+ cells to die. The treatment also prevented a layer of connective tissue, known as the endometrial stroma, from forming in the uterus. As a result, the mice and rats were infertile and had severely underdeveloped uteri.

While the Misr2+ cells are present in newborn rats and mice, Saatcioglu et al. found that they disappeared before birth in humans. However, the overall results suggest that Misr2+ cells act as progenitor cells that develop into the cells of the endometrial stroma. Future work could investigate the roles these cells play in causing uterine developmental disorders and infertility disorders. Furthermore, the finding that MIS inhibits the Misr2+ cells could help researchers to develop treatments for uterine cancer and other conditions where the cells of the uterus grow and divide too much.

DOI: https://doi.org/10.7554/eLife.46349.002

Amhr2) pathway is highly specific to this process, since either ligand or receptor knockout mice present with identical phenotypes of persistent Mullerian duct syndrome (PMDS), a rare form of male pseudohermaphroditism in mice (*Behringer et al., 1994*; *Mishina et al., 1996*; *Mishina et al., 1999*) and humans (*Imbeaud et al., 1994*). It is thought that the narrow developmental window when Mullerian duct regression occurs is the only period when these Misr2+ mesenchymal cells are able to respond to MIS, beyond which further differentiation of the duct renders it insensitive to this inhibitory signal (*Josso et al., 1976*). Indeed, transgenic mice with constitutive overexpression of MIS, driven by a metallothionein-1 promoter, during the critical period of Mullerian duct formation displayed Mullerian agenesis and quickly lost all germ cells in the ovary after birth (*Behringer et al., 1990*).

In females, which do not express MIS during embryonic development, the fate of these subluminal Misr2+ mesenchymal cells remains unclear. In adult mice, expression of Misr2 is restricted to the myometrium, suggesting a common origin for both cell types (*Arango et al., 2005*; *Arango et al., 2008*); however, the lack of an inducible Misr2 reporter mouse has precluded a precise lineage tracing. Many cell types of the urogenital ridge primordium including the coelomic epithelium, mesonephric mesenchyme, and Wolffian duct epithelium are thought to contribute to the formation and development of the Mullerian duct as it further differentiates into the oviduct, uterus, cervix, and upper vagina (*Fujino et al., 2009*; *Mullen and Behringer, 2014*). In rodents, most of the differentiation of the uterine layers occurs postnatally; at birth, the uterus consists of a single layer of luminal epithelium surrounded by undifferentiated mesenchyme which later develops into myometrium and endometrial stromal cells, while the epithelium subsequently gains the ability to form endometrial glands though invagination at postnatal day (PND) 6–9 (*Branham et al., 1985*; *Brody and Cunha, 1989*). The timing of development of the endometrial stroma, and its contribution to the coordination of development of the myometrium and endometrial glands in early postnatal development are poorly understood. Mutations of genes which orchestrate early Mullerian mesenchyme development can have drastic consequences on female fertility and lead to Mullerian aplasia or uterine hypoplasia as observed in the Mayer-Rokitansky-Kuster-Hauser syndrome (*Patnaik et al., 2015*). Therefore, the pathways involved in early postnatal specification of the uterine compartments are critical to our understanding of Mullerian development and uterine factor infertility. Here, we characterize the persistence of subluminal Misr2+mesenchymal cells beyond the embryonic period of sexual

differentiation, document their retained sensitivity to MIS neonatally, and characterize their critical role in the development of the endometrial stroma.

## Results

### Female mullerian subluminal mesenchyme maintains Misr2 expression and MIS sensitivity postnatally in rodents

To determine the fate of the Mullerian subluminal mesenchyme in the developing uterus, we sought to identify specific markers whose expression in that cell type perdured postnatally. Although the embryonic development of the *Misr2+* Mullerian mesenchyme has been extensively studied (*Jamin et al., 2002*; *Arango et al., 2008*; *Kobayashi et al., 2011*), its early postnatal fate has not. Using lineage tracing in a Misr2-CRE/TdTomato reporter transgenic cross in C57BL/6 mice, we first confirmed that embryonic urogenital *Misr2+* intermediate mesoderm gives rise to both the endometrial and the myometrial layers of the uterus, but not its epithelium (*Figure 1—figure supplement 1A*). Because Misr2-CRE is not inducible, any Misr2 expression during early development will result in permanent expression of the TdTomato reporter (*Figure 1—figure supplement 1A*) Therefore, to track further the *Misr2+* cells in Mullerian mesenchyme, we conducted careful spatiotemporal studies using *Misr2* RNA in situ hybridization (RNAish) from the embryonic period (E14-15) into postnatal life (*Figure 1A*). As expected, expression of *Misr2* is restricted to the mesenchyme surrounding the Mullerian duct in both male and female urogenital ridges during embryonic development (E17-19) (*Figure 1A*). Postnatally, *Misr2* expression becomes increasingly restricted to a thin band of subluminal mesenchyme, while being excluded from the epithelium and developing myometrium (*Figure 1A*, PND 0, PND 2) (*Figure 1A*, *Figure 1—source data 1*). Following differentiation of the functional layers of the uterus around PND 6 (*Brody and Cunha, 1989*), *Misr2* expression commences to be detectable in the myometrium consistent with previous findings (*Arango et al., 2008*) (*Figure 1A*).

To evaluate the effect of MIS on the development of the uterus, we chose to use rats which have larger litters and display a similar spatiotemporal pattern of *Misr2* expression (*Figure 1—figure supplement 1D–E*, *Figure 1—source data 1*). Furthermore, in rats, *Misr2* expression is gradually attenuated in the PND1-6 period, both at the proximal (cervix) and at the distal (oviduct) ends of the developing uterine horns (*Figure 1—figure supplement 1D*) coinciding with the timing of expansion of the endometrial stroma, and the rise in secretion of MIS from the developing ovaries, which was measured in the rat serum by ELISA (*Figure 1—figure supplement 1B and C*). To evaluate the sensitivity of *Misr2+* mesenchymal cells to MIS during this postnatal period, we treated rat pups with adeno-associated viral vectors (AAV9) (*Pépin et al., 2015*; *Kano et al., 2017*) delivering MIS at PND 1 (*Figure 1B* and *Figure 2A*). Administration of a single dose of AAV9-MIS (5E10 particles/pup) on PND 1 led to a robust induction (2.8 ± 0.7 µg/ml) of circulating exogenous MIS as measured by ELISA on PND6 (*Figure 2B*). Postnatal exposure to MIS led to an alteration in the appearance of the uterus by PND 6, and severe uterine hypoplasia by PND 20, suggestive of a failure of the uterus to develop beyond its perinatal state (*Figure 2C*). Histomorphological analyses of transverse uterine sections revealed smaller uteri with underdeveloped endometrial stromal layers and smaller lumina in MIS-treated rats compared to their sibling controls at all time points analyzed (PND 6, 10, and 20) (*Figure 2C and D*, *Figure 2—source data 1*).

Treatment with MIS prevented the gradual decrease of *Misr2+* cells, and inhibited the expansion of the endometrial stroma normally observed by PND 6 (*Figure 2E*). For consistency and clarity, we will refer to these persistent *Misr2+* subluminal mesenchymal cells in the MIS-treated animal as 'inhibited progenitors', although this function remains presumptive. In response to MIS, these inhibited putative progenitors expressed high levels of *Smad6*, a negative regulator of TGFβ signaling, and a previously reported canonical downstream target of MIS in the Mullerian mesenchyme (*Figure 2E*), suggesting MIS signaling was both operational and autonomous to these *Misr2+* cells (*Clarke et al., 2001*; *Mullen et al., 2018*).

Interestingly, the endometrial stromal hypoplasia resulting from MIS exposure precluded later development of endometrial glands, as confirmed by absence of Foxa2 marker expression in immunofluorescence and qPCR (*Figure 2—figure supplement 1A and B*). However, the development of the circular myometrial layers was unaffected by exposure to MIS, as shown in the MIS-treated

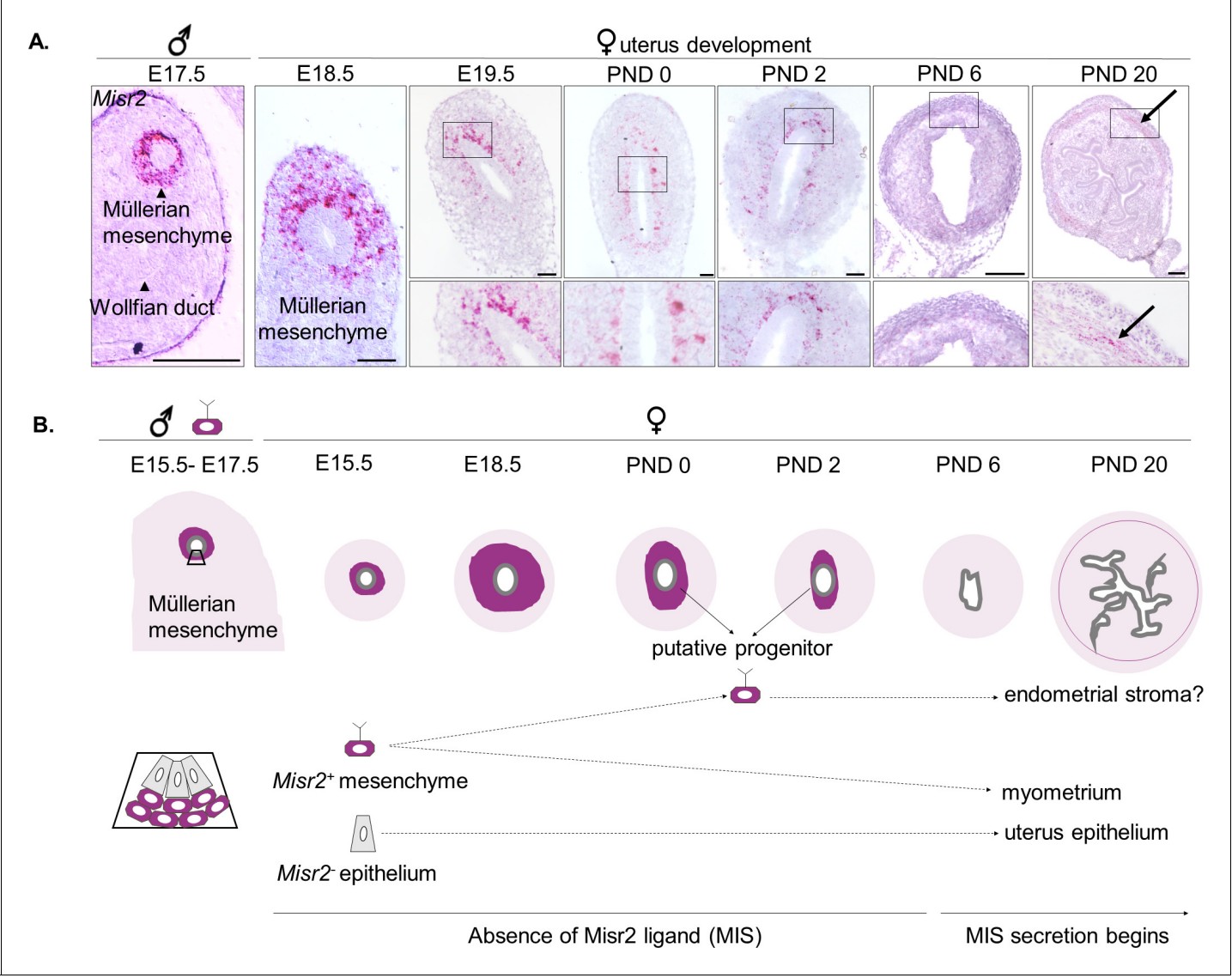

**Figure 1.** Subluminal mesenchymal cells of postnatal uteri retain expression of *Misr2*. (**A**) RNAish (RNA scope) analysis of *Misr2* in transverse sections of male urogenital ridges at E 17.5, and in a time series of the developing uteri including E 18.5, E 19.5, PND 0, 2, and 6 in mice). Scale bars = 50 µm (n = 8 for<PND6; n = 4 for>PND6). Number of mice analyzed per time point is presented in *Figure 1—source data 1*. Black arrows demarcate to the myometrial layer at PND 20. (**B**) Representative scheme of the Misr2 expression pattern in the developing uterus. Subluminal mesenchymal cells continue to express *Misr2* in the postnally until around PND 6. We sought to investigate the fate of these postnatal *Misr2+* subluminal cells, and their possible role as progenitor cells of the endometrial stroma.

DOI: https://doi.org/10.7554/eLife.46349.003

The following source data and figure supplement are available for figure 1:

**Source data 1.** Number of replicates per time point for the *Misr2* in situ analysis in mice (*Figure 1A*) and in rats (*Figure 1—figure supplement 1D–E*).
DOI: https://doi.org/10.7554/eLife.46349.005

**Figure supplement 1.** *Misr2* expression is gradually attenuated in the PND 1-6 period, coinciding with the timing of the rise in secretion of MIS.
DOI: https://doi.org/10.7554/eLife.46349.004

transverse uterine sections by immunofluorescence, and qPCR analysis of smooth muscle markers such as smooth muscle actin (Acta2) and transgelin (Tagln), which remain similar at various time points between control and MIS exposed uteri (*Figure 2—figure supplement 1C and D*). Surprisingly, the inhibited progenitor cells expressed Acta2 ectopically (shown in red) as well as Vimentin (shown in green) in response to MIS on PND 6 (*Figure 2F*), suggesting a metaplastic effect of MIS.

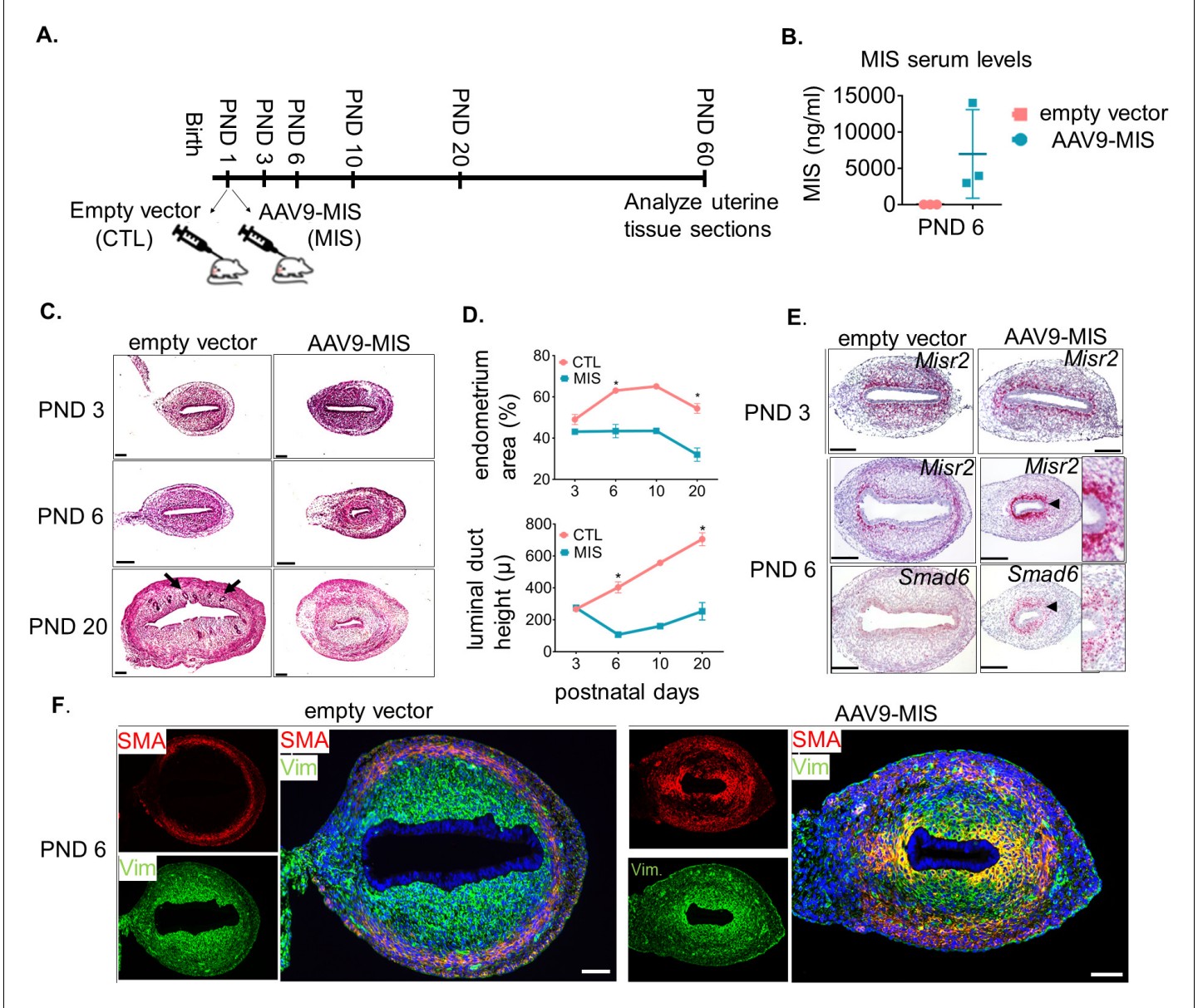

**Figure 2.** *Misr2+* subluminal mesenchymal cells are susceptible to inhibition by MIS. (**A**) Rat pups were treated with AAV9-MIS (MIS) or empty vector control (CTL) on postnatal day1 (PND 1) and euthanized at different developmental time points (**B–F**). Rats were used as the initial model organism since their litter sizes are bigger than mice. (**B**) MIS serum levels from control and AAV9-MIS treated rats on day 6 (n = 3 for both). (**C**) H& E sections from CTL and AAV9- MIS treated uteri on PND 3, 6, and 20. Endometrial glands are demarcated by black arrows on day 20. Scale bars = 100 μm. (**D**) Percentage of the endometrial stroma area (%), and luminal duct height of the CTL and MIS-treated uteri (*Figure 2—source data 1*). (**E**) *Misr2* and *Smad6* expression pattern by RNAish. Scale bars = 100 μm. (**F**) Smooth muscle α-actin (SMA) in red, and Vimentin (Vim) in green on CTL and AAV9-MIS treated uterine sections (PND 6, scale bars = 100 μm).

DOI: https://doi.org/10.7554/eLife.46349.006

The following source data and figure supplement are available for figure 2:

**Source data 1.** Data, number of replicates and p values of significance between the control and AAV9-MIS treated uterine samples for histomorphological analysis.

DOI: https://doi.org/10.7554/eLife.46349.008

**Figure supplement 1.** MIS treatment inhibits endometrial gland formation but not myometrial development.

DOI: https://doi.org/10.7554/eLife.46349.007

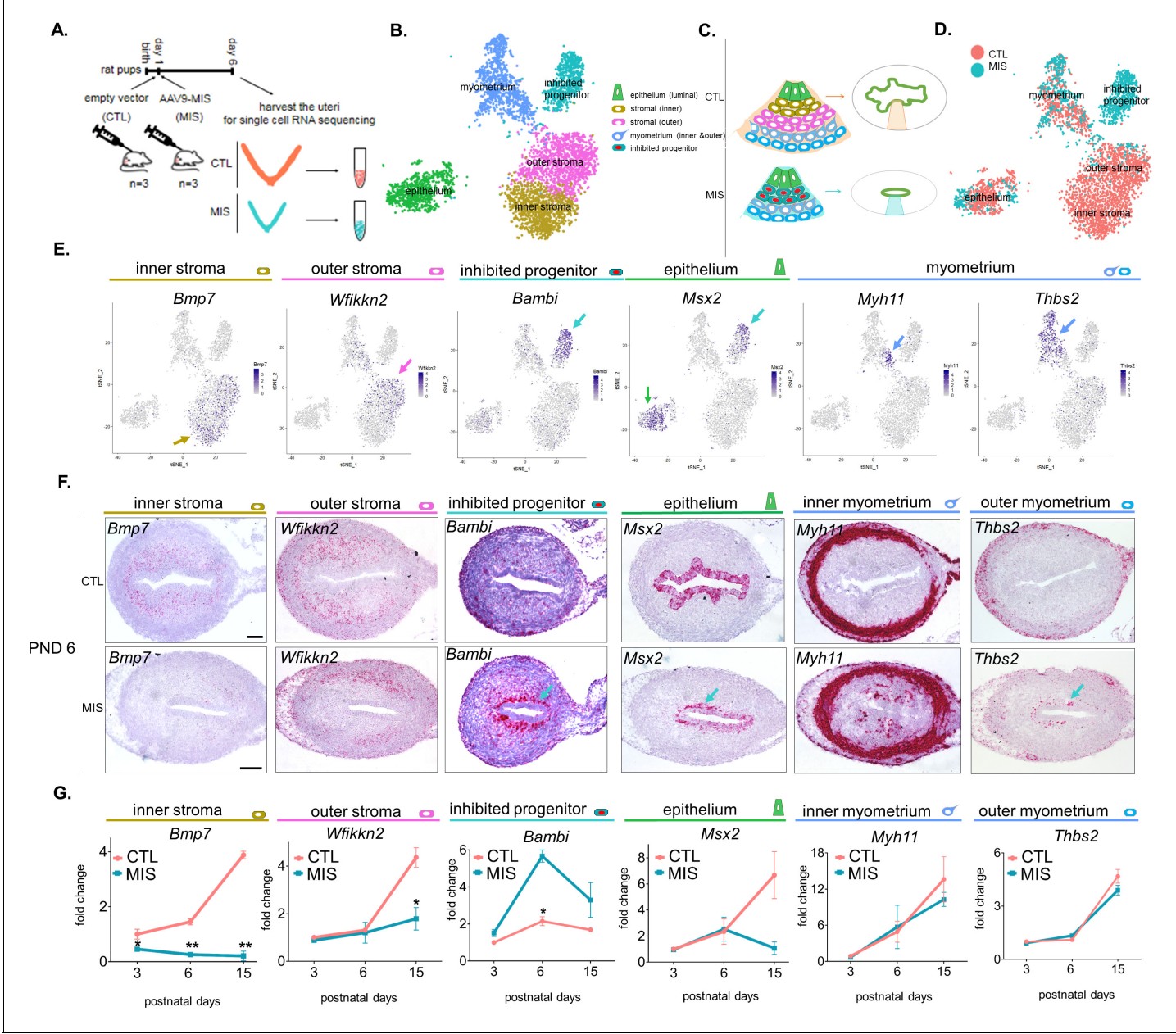

**Figure 3.** Single cell RNA sequencing of PND 6 uteri reveals distinct cell types and unique gene signatures in response to MIS treatment. (**A**) Rat pups were treated with empty vector control (CTL) or AAV9-MIS (MIS) on PND 1, and euthanized on PND 6 (n = 3 for both). Following whole-tissue dissociation, RNA isolated from single uterine cells, were barcoded and sequenced using inDROP. (**B**) t-SNE plot of unbiased clustering of uterine cells, where each color-coded cluster represents one cell type/state (only the main uterine parenchymal clusters are represented) (*Figure 3—source data 1*). (**C**) Schematic representation of the differential cellular composition of control and MIS-treated uteri. (**D**) t-SNE plot of unbiased clustering of uterine cells (dots) color-coded by treatment with CTL (orange) and MIS (blue) (*Figure 3—source data 1*). (**E**) Gene expression levels of representative cell-type-specific markers for each cluster overlaid on t-SNE plots (featureplot, color-coded arrow refers to cell type). (**F**) RNAish stains of representative cell-type markers in transverse uterine sections of empty vector control (CTL) and AAV9-MIS (MIS) treated mice at PND 6. Scale bars = 100 μm, same for all sections. Mouse tissues were used for validation purposes as the RNA in situ probes were readily available for mice and the effect of MIS was conserved among mice and rats. (**G**) Rat pups treated with AAV9-MIS (MIS) or empty vector (CTL) on PND 1 were euthanized at different developmental time points (PND3, 6, and 15), and their uteri were harvested for QPCR analysis of one representative marker for each cluster (n > 2, unpaired Student's t test, mean ± SEM, *(p≤0.05), **(p≤0.01)) (*Figure 3—source data 3*).

DOI: https://doi.org/10.7554/eLife.46349.009

The following source data and figure supplements are available for figure 3:

**Source data 1.** Related to *Figure 3*; *Figure 3—figure supplements 1*, *2*, *3* and *4*; *Figure 4*, *Figure 4—figure supplements 1*, *2* and *3* and *Table 1*.

*Figure 3 continued*

DOI: https://doi.org/10.7554/eLife.46349.014
**Source data 2.** Differentially expressed genes (MIS vs Control) in the myometrium of the developing rat uteri.
DOI: https://doi.org/10.7554/eLife.46349.015
**Source data 3.** Related to *Figure 3G*, *Figure 4—figure supplements 1D,3D,4E*.
DOI: https://doi.org/10.7554/eLife.46349.016
**Figure supplement 1.** Single-cell RNA sequencing analysis of the control and AAV9-MIS treated rat uterine cells.
DOI: https://doi.org/10.7554/eLife.46349.010
**Figure supplement 2.** Single-cell RNA-seq reveals the cell atlas of the developing PND6 uterus.
DOI: https://doi.org/10.7554/eLife.46349.011
**Figure supplement 3.** Single-cell RNA-seq reveals the cell atlas of the developing PND 6 uterus.
DOI: https://doi.org/10.7554/eLife.46349.012
**Figure supplement 4.** The myometrium cluster can be sub-divided into three different cell types whose gene signatures are only weakly affected by MIS treatment.
DOI: https://doi.org/10.7554/eLife.46349.013

## Single Cell RNA sequencing of the PND 6 uterus uncovered an unexpected diversity of cell types in control and MIS-treated uteri

To understand how MIS dysregulates stromal differentiation, we followed the fate of the *Misr2+* subbluminal mesenchymal cells following MIS treatment using single-cell RNA sequencing (scRNAseq), based on droplet microfluidic sorting of single cells (inDROP) (*Klein et al., 2015*). We performed an enzymatic digestion of the whole uteri on PND 6 (*Figure 3A*), a time point at which the *Misr2*$^+$ subluminal mesenchymal cells are almost completely gone in the normal uterus, but are retained in the MIS-treated ones (*Figure 2E*).

The uterine horns of three control (treated with 5E10 empty vector particles/pup on PND1) and three AAV9-MIS-treated animals (treated with 5E10 particles/pup on PND1) were recovered at PND 6 and digested in a cocktail of proteases into single-cell suspensions (*Figure 3A*). The inDROP libraries were sequenced, demultiplexed, normalized, and analyzed using the Seurat package in 'R', as previously described (*Butler et al., 2018*). The processed sample of 9801 cells clustered into 15 groups: outer stroma (1494), inner stroma (1081), myometrium (1172), inhibited progenitor (946), luminal epithelium (898), dividing mesenchyme (750), vascular endothelium (535), dividing epithelium (314), mesothelial (313), myeloid (297), erythroid (155), pericyte (139), lymphatic endothelium (54), and nerve cells (33), along with a low-information cluster (high % of mitochondrial genes) which was censored from further analysis (1620) (*Figure 3—figure supplement 1A–C*, *Figure 3—source data 1*). Cell identity of each cluster was assigned based on the presence of previously reported cell-type markers using the top 15 significantly enriched genes (by adjusted p-value) of the single cell analysis, summarized in *Table 1*. Markers include *Smoc2* (*Mucenski et al., 2019*) for stromal cells; *Ptn* (*Mucenski et al., 2019*) for myometrial cells; *Epcam* and *Wnt7a* (*Litvinov et al., 1997*) (*Miller and Sassoon, 1998*) for luminal epithelium cells; *Top2a* (*Whitfield et al., 2006*) for proliferating cells, *Cdh5* (*Bhasin et al., 2010*) for vascular endothelial cells; *Wnt7a* and *Hmgb2* for dividing epithelium (*Miller and Sassoon, 1998*) (*Stros et al., 2009*); *Msln* for mesothelium (*Chang and Pastan, 1996*); *Csf1r* (*MacDonald et al., 2010*) for myeloid cells (*Zilionis et al., 2019*); *Alas2* (*Kaneko et al., 2014*) for erythroid cells; *Mcam* (*Barron et al., 2016*) for pericytes; *Lyve1* (*Hirakawa et al., 2003*) for lymphatic endothelium; and *Sox10* (*Nonaka et al., 2008*) (*Zeisel et al., 2018*) for neuronal cells (*Table 1*). A heatmap of the top five genes for each cluster (by fold expression over average) is also presented in (*Figure 3—figure supplement 2*). The top two markers generated from the heatmaps were also analyzed by violin plots among 14 different clusters (*Figure 3—figure supplement 3*).

Subsequent analyses focused on the five clusters representing the functional layers of the PND 6 uterus including luminal epithelium, myometrium, inner and outer endometrial stroma, and a unique cluster of cells responding to MIS treatment (inhibited progenitor) (*Figure 3B and C*). To validate the gene signatures of these clusters, and their changes in response to MIS treatment, we selected one marker gene (*Figure 3—figure supplement 1D*) for each unique cluster, and confirmed its cell-type-specific expression pattern in transverse uterine tissue sections of control as well as treated mice uteri at PND 6 by RNAish (*Figure 3E and F*). Endometrial stromal cells were found to have two transcriptionally distinct cell types: inner stroma (*Bmp7+*) and outer stroma (*Wfikkn2+*) (*Figure 3B*,

**Table 1.** Unique gene signatures of the 14 different cell types (clusters) from the developing rat uteri on PND 6. Gene markers revealed by single cell RNA sequencing, sorted by highest adjusted p-value with positive LogFc (*Figure 3—source data 1*). Clusters identities were assigned to different cell types by RNAish validation and/or manual literature review. Top 15 markers, citations, and associations are listed in the table. Selected markers were further validated in *Figure 3*.

| Outer stroma (1) | Myometrium (2) | Inner stroma (3) | Inhibited Progenitor (4) | Epithelium (5) | Proliferating cells (6) | Vascular endothelium (7) | Dividing epithelium (8) | Mesothelium (9) | Myeloid (10) | Erythroid (11) | Pericyte (12) | Lymphatic endothelium (13) | Neuronal (14) |
|---|---|---|---|---|---|---|---|---|---|---|---|---|---|
| RGD1305645 | Lum | Nrgn | AC109891.2 | Cd24 ** | Hmgb2 | Cd93 * | Epcam * (Litvinov et al., 1997) | Bsg | Fcer1g * | Hba.a1 * | Mgp * | Ccl21 | Plp1 * |
| Smoc2 * (Mucenski et al., 2019) | Postn | Cpxm2 ** | Rn60_1_221 6.1 | Epcam * (Litvinov et al., 1997) | Stmn1 * | Cdh5 ** (Bhasin et al., 2010) | Cldn3 | Upk1b * | Tyrobp * | Hba.a2 * | Mcam ** (Barron et al., 2016) | Mmrn1 ** | Sox10 * (Nonaka et al., 2008) |
| Dpt * | NEWGENE_621351 | Tgfbi * | AABR07068705.1 | Wfdc2 * (Mucenski et al., 2019) | Top2a * (Whitfield et al., 2006) | RGD1310587 | Cd24 * | Msln ** (Chang and Pastan, 1996) | Csf1r ** (MacDonald et al., 2010) | LOC1036948 57 | Rergl * | Lyve1** (Hirakawa et al., 2003) | Gfra3 * |
| Cpe | Col1a1 | Plac8 | Fgfr2 | Wnt7a * (Miller and Sassoon, 1998) | Mki67 * (Whitfield et al., 2006) | Emcn * | Cdh1 * | Fmod | Lyz2 * | Hbb * | Eln * | Flt4 ** (Hirakawa et al., 2003) | Abca8a |
| Vcan * | Ptn * (Mucenski et al., 2019) | Vcan * | Unc5b * (Mullen et al., 2018) | Cldn3 | Tubb5 * | Tie1 * | Klf5 | Cav1 | Ftl1 | LOC1036948 55 | Col4a1 * | Cldn5 * | Egfl8 |
| Dcn * | Thbs2 ** | Fn1 * | Ugt1a2 | Krt8 * | LOC1003595 39 | Plvap | Krt8 * | Dpp4 * | Cybb * | Alas2 * (Kaneko et al., 2014) | Igfbp7 * | Fgl2 * | Cdh19 |
| Col4a5 * | Col3a1 | Apcdd1 | Bambi ** (Mullen et al., 2018) | Cdh1 * | Racgap1 | Cav1 | Mt1 | Anxa3 | Laptm5 * | Ybx3 * | Cspg4 ** | Klhl4 * | Afap1l2 |
| Apoe | Ogn (Mucenski et al., 2019) | Cdh11 * | Srgn | Tacstd2 | LOC1003603 16 | Adgrl4 | Wnt7a * (Miller and Sassoon, 1998) | Igfbp6 | Tmsb4x | Hba.a3 * | Abcc9 * | Cdh5 * | L1cam * |
| Osr2 | Sparc | Col6a3 * | Kcnk3 | Cldn4 * | Prc1 * | Plxnd1 | Mt2A | Cfb | Ctsb * | Alox15 | Epas1 | Tbx1 | Col5a3 |
| Tnfrsf21 | Cxcl12 | Tnfrsf21 | Igfbp5 | Msx1 | Cdk1 * | Epas1 * | Msx1 | Lox | Lcp1 * | Car2 | Rgs5 | Slc45a3 | Olfml2a |
| Adamts7 | Igfbp5 | Nkd2 | Igf2r | Klf5 | Depdc1 | Esam * | Wfdc2 | Muc16 * | Ptprc * | Ahsp * | Foxs1 * | Adgrg3 | Metrn |
| Slc26a7 | Ccdc80 | Axin2 | Plac8 | Elf3 * | Smc2 | Cyyr1 | Dlx5 | Itm2a | C1qa * | Rbm38 | Myl9 | Sdpr | Mpz * |
| Wfikkn2 ** | Gpc3 | Vim * | Prrx2 | Dlx5 | Klf11 * | Clec14a | Sbspon | Sema3c | C1qc * | Lgals5 | RGD1 564664 | LOC1009120 34 | Plekha4 |
| Col6a3 * | Col6a2 * | Alpl | | Aldh2 | Cenpf * | Adgrf5 | Hmgb2 ** (Stros et al., 2009) | Fbln2 * | Aif1 * | Slc4a1 | Acta2 * | Rn50_1_043 5.2 | Dbi |
| Islr | Itm2a * | | AAB R070253 16.1 | Mt2 | Tmpo * | Col3a1 | Tacstd2 | Eln | Cxcl2 | Slc25a39 | Cald1 * | Il2rg | Col20a1 |

*Table 1 continued on next page*

*Table 1 continued*

| Outer stroma (1) | Myometrium (2) | Inner stroma (3) | Inhibited Progenitor (4) | Epithelium (5) | Proliferating cells (6) | Vascular endothelium (7) | Dividing epithelium (8) | Mesothelium (9) | Myeloid (10) | Erythroid (11) | Pericyte (12) | Lymphatic endothelium (13) | Neuronal (14) |
|---|---|---|---|---|---|---|---|---|---|---|---|---|---|
| * extracellular matrix, collagen or cytoskeleton related **validated new marker (in uterus) | * muscle related **validated new marker (in uterus) | *extracellular matrix, collagen or cytoskeleton related **validated new marker (in uterus) | *implicated during Mullerian duct regression in males ** validated new marker (in uterus) | *epithelium associated **validated new marker (in uterus) | *cell cycle or proliferation associated | *endothelial related **implicated higher expression in vascular tissues | *epithelium associated **cell proliferation associated | *mesothelium associated or enriched **mesothelium specific | * myeloid associated (*Zilionis et al., 2019*) **myeloid specific | *hemoglobin associated | * fibroblast associated **pericyte marker | *endothelial related ** lymphatic endothelium associated | *neuronal related (*Zeisel et al., 2018*) |

DOI: https://doi.org/10.7554/eLife.46349.024

*C*). Further subdivision of the myometrium cluster led to three distinct cell subtypes in the developing myometrial layer, all of which presented similar numbers of control and treated cells: inner myometrium (*Myh11+*), outer myometrium (*Thbs2+*), and interstitial (myofibril enriched) myometrium (*Mfap5+*) (*Figure 3D–F*, *Figure 3—figure supplement 4*, *Figure 3—source data 2*).

## MIS blocks the expansion and differentiation of an *Misr2+* putative stromal progenitor

We next focused on differential gene expression in response to MIS treatment using our dataset of 6811 control and 2990 MIS-treated cells. In our cell atlas, the control and treated uterine cell datasets were of similar quality, and had comparable distributions of unique molecular identifiers (UMI) and gene numbers (*Figure 3D*, *Figure 3—figure supplement 1B*, *Figure 3A*). While both the epithelial (CTL 9.63%, MIS 8.06%) and the myometrial (CTL 11.98%, MIS 16.72%) (*Figure 3—figure supplement 1B–C*, *Figure 3—figure supplement 4*) cell types were re-presented in similar proportions in both datasets, 'inner' (CTL 18.68%, MIS 3.28%) and 'outer' (CTL 15.64%, MIS 0.27%) endometrial stroma clusters (*Figure 3—figure supplement 1B–C*) were almost entirely composed of control cells unlike the 'inhibited progenitor' cluster, which was unique to the MIS treatment (*Figure 3B–D*, *Figure 3—figure supplement 1B–C*, *Figure 4A*, *Figure 4—figure supplement 1*). *Misr2* expression was enriched only in the inhibited progenitor cluster (*Figure 3C–D*, *Figure 4A*) as previously observed by RNAish (*Figure 2E*). MIS stimulation caused the sub-luminal inhibited progenitor cells to persist longer and induced overexpression of genes related to the bone morphogenic protein and transforming growth factor beta (BMP/TGFβ) signaling pathways (*Smad6, Bambi*), epigenetic markers (*Hdac4*), and, strikingly, to express ectopically smooth muscle (*Thbs2*) and epithelial (*Msx2*) markers, reflecting the multipotent characteristics of the early Mullerian progenitors (*Figure 3E–G*, *Figure 4A*, *Figure 4—figure supplement 1A–D*, *Table 1*). QPCR analysis of these ectopically expressed genes revealed that the large differences observed at PND 6 diminished with time, and became less pronounced by PND 15 (*Figure 3G*, *Figure 4A*, *Figure 4—figure supplement 1*; *Figure 4—figure supplement 2*; *Figure 4—figure supplement 1B*, *Figure 3—source data 3*).

Cleaved caspase-3 histological staining of the MIS-treated uteri confirmed that the inhibited progenitor cells underwent apoptosis at approximately 9 days after treatment, whereas normally developing endometrial stromal cells were negative for the apoptotic marker (*Figure 4—figure supplement 1E*). QPCR and *RNAish* validation of differentially expressed genes in the stroma confirmed that both *Wfikkn2* ('outer stroma'), *Bmp7* ('inner stroma') failed to be induced over time in the MIS treated group (*Figure 3G*), along with other markers such as *Cpxm2* and *Enpp2* (*Figure 4—figure supplement 3A–D*), consistent with the hypothesis that MIS prevented the subluminal progenitor cells from undergoing stromal specification and amplification; instead they eventually underwent apoptosis (*Figure 4—figure supplement 1E*).

## MIS disrupts gene expression in the luminal epithelium in a non-cell-autonomous manner

Our results indicate that neonatal exposure to MIS induces changes in gene expression in the epithelial cell cluster as visible by the shifted treatment population in the t-SNE plot (*Figure 3D*, *Figure 3—figure supplement 1B*, *Figure 4—figure supplement 4A*). Even though expression of the basic luminal epithelial cell markers were not significantly different between control and MIS-treated cells (*Ecad, Cd24,* and *Klf5*) (*Figure 4—figure supplement 4* (A-E), differential gene expression analysis of the epithelial cluster based on treatment (CTL and MIS) revealed gene candidates whose expressionwas significantly changed by MIS treatment (*Figure 4—figure supplement 4F–I*) despite the lack of *Misr2* expression in epithelial cells (*Figure 1A,E*). We confirmed that *Id3* was downregulated in the MIS-treated epithelial cells (*Figure 4—figure supplement 4H*). Another intriguing gene expression pattern in response to MIS treatment was the downregulation of the epithelial cell marker *Msx2* at PND6, which coincides with its ectopic expression in the inhibited putative progenitor (*Figure 3F–G* (Msx2), *Figure 4A*, *Figure 4—figure supplement 4F*). These results are consistent with MIS mediating the indirect repression of epithelial *Msx2* and *Id3* through paracrine signals emanating from the *Misr2+* mesenchymal cells, or the lack of normal endometrial stromal signals, which

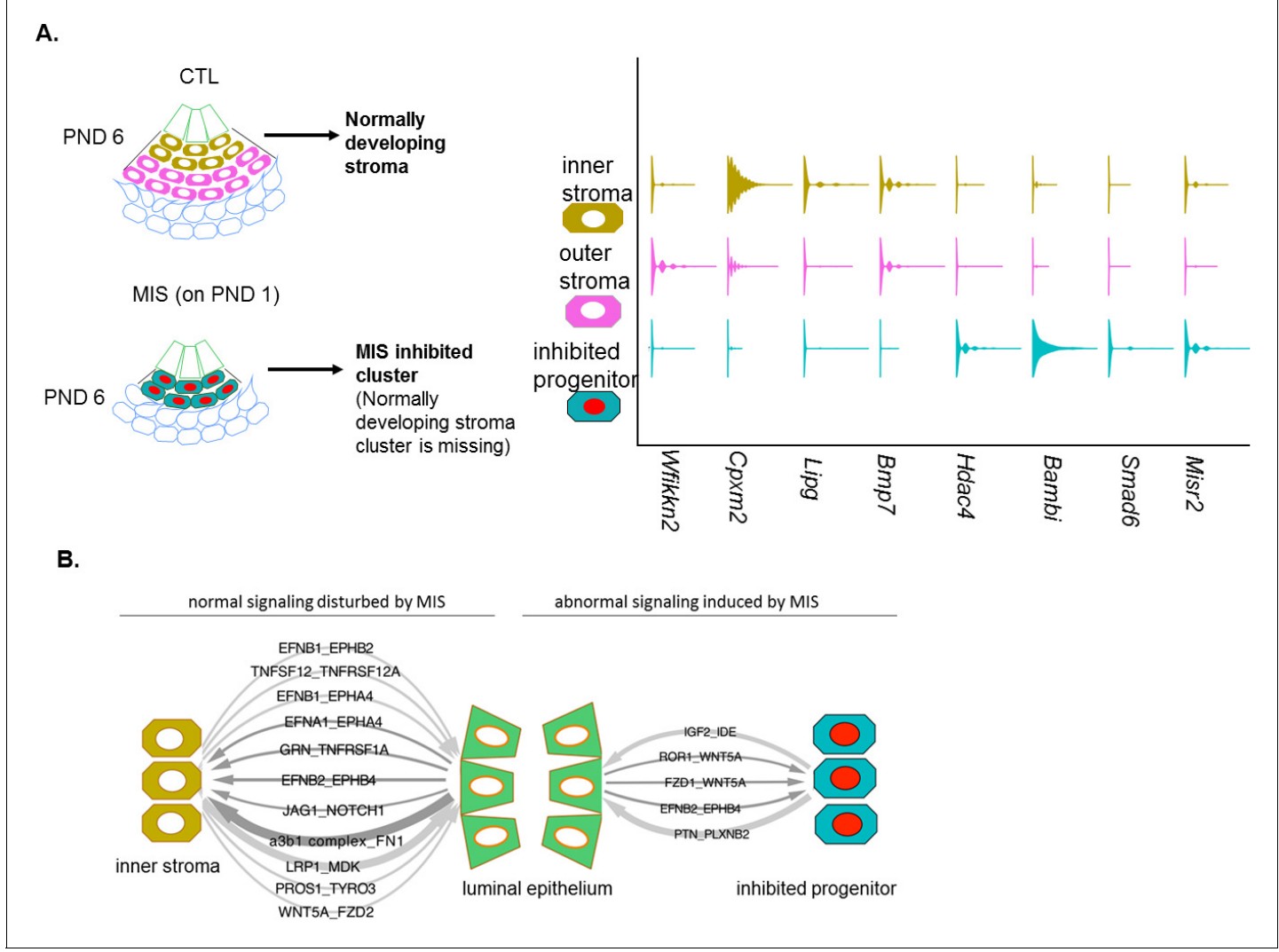

**Figure 4.** MIS treatment blocks the differentiation of a putative progenitor cell, preventing endometrial stroma formation, which indirectly dysregulates epithelial cell development. (**A**) When treated with MIS, a unique population of *Misr2+* subluminal putative progenitor cells are retained (inhibited progenitors), while the inner and the outer endometrial stromal layers fail to develop (representative scheme on the left). Note the differential expression pattern of stromal markers in the violin plots of the control and the MIS-treated uterine cells. See *Figure 3* and *Figure 4—figure supplements 1–3* for validation of these markers on tissue sections. (**B**) Diagrams of the unique receptor-ligand pairs present 1) between luminal epithelial cells and inner stromal cells (which are absent in MIS-treated uteri) (left), and 2) between the epithelial cells and the inhibited progenitor cells, (which are only present in the MIS-treated group).

DOI: https://doi.org/10.7554/eLife.46349.017

The following source data and figure supplements are available for figure 4:

**Source data 1.** Cell phone DB analysis.
DOI: https://doi.org/10.7554/eLife.46349.022

**Source data 2.** Differentially expressed genes (MIS vs Control) in the luminal epithelium of the developing rat uteri.
DOI: https://doi.org/10.7554/eLife.46349.023

**Figure supplement 1.** The inhibited progenitor cluster is only present in the MIS-treated uterus and corresponds to a subluminal mesenchymal cell.
DOI: https://doi.org/10.7554/eLife.46349.018

**Figure supplement 2.** Endometrial stromal layers fail to form when the putative progenitor cells are inhibited by MIS.
DOI: https://doi.org/10.7554/eLife.46349.019

**Figure supplement 3.** Single-cell RNA sequencing reveals the gene signature of uterine epithelial cells and a non-cell autonomous response to MIS.
DOI: https://doi.org/10.7554/eLife.46349.020

**Figure supplement 4.** Receptor-ligand interactions in the control and MIS-treated uterine atlases.
DOI: https://doi.org/10.7554/eLife.46349.021

in turn may prevent subsequent endometrial gland formation at later time points (*Figure 4—figure supplement 4I*, *Figure 2—figure supplement 1A–D*).

Finally, to survey the paracrine signaling between the inner stroma and epithelium and catalog how it may be disrupted by MIS treatment (inhibited putative progenitor and epithelium), we performed a comprehensive ligand/receptor analysis using the CellPhoneDB algorithm (*Vento-Tormo et al., 2018*) (*Figure 4B*- *Figure 4—figure supplement 4*, *Figure 4—source data 1*). Briefly, significantly expressed ligand/receptor pairs were systematically cataloged between each functional cell types (*Figure 4B*), revealing important developmental pathways dysregulated in MIS-treated uteri, such as Wnt and Igf2 signaling (*Figure 4B*, *Figure 4—figure supplement 4*, *Figure 4—source data 1*).

## Misr2+ putative progenitors are necessary for uterine development and fertility only during the first 6 days of life

The timing of expression of *Misr2* in the subluminal mesenchyme led us to hypothesize that limiting exposure to MIS during only the first 6 days of development could be sufficient to explain the observed long-term uterine hypoplasia. To test this hypothesis, we treated rat pups with recombinant MIS protein (rhMIS) during 6 days intervals (3 mg/kg) starting from days 1, 6, or 11 (*Figure 5A*). Consistent with this hypothesis, only when rats were treated during PND1-6 period were the uteri smaller than controls, at PND6, 20, 45, and even up to 8 months (*Figure 5B,C,G*). In contrast, when rhMIS was administered starting from PND6 or PND11, uterine hypoplasia was muted or absent, suggesting a narrow window of susceptibility with long-lasting consequences (*Figure 5B*). Therefore, only perinatal (PND1-6) administration of rhMIS completely phenocopied the continuous exposure phenotypes observed with AAV9-MIS, including lower percentage of endometrial stromal cells, smaller luminal ducts, and absence of glandular development (*Figure 5B*, *Figure 5—figure supplement 1A*) as confirmed by downregulation of Foxa2 expression and absence of glandular ducts (*Figure 5D–E*). Strikingly, this 6-day postnatal treatment was also sufficient to cause complete infertility later in adulthood (n = 3 per group, p≤0.05) (*Figure 5F*). Analysis of the control and the 6-day rhMIS-treated uteri at a later time point (8 months) confirmed that the profound endometrial stromal hypoplasia persists in the adult (*Figure 5G–H*). In contrast, the ovaries fully recover from the short MIS inhibition of folliculogenesis (PND1-6) and display normal ovarian sizes and follicular composition at this timepoint (*Figure 5H*).

The impaired uterine development and infertility is unlikely to be secondary to ovarian suppression since folliculogenesis only starts reaching early pre-antral stages by PND6, and MIS treatment appears to have little effect on steroid hormones (E2 and P4) during that time (*Figure 5—figure supplement 1B*). Furthermore, as previously described (*Kano et al., 2017*), MIS inhibition of folliculogenesis by rhMIS is reversible, and while the MIS-treated PND1-6 ovaries showed an initial delay in folliculogenesis at early timepoints (PND 20), it was resolved by PND 45 confirming no lasting impact on ovarian function (*Figure 5—figure supplement 1C–D*).

To confirm that the effect of MIS on the *Misr2+* putative stromal progenitor was intrinsic to the uterus (and not the ovary), we treated gonadectomized rat pups with control or AAV9-MIS on PND2 (*Figure 5—figure supplement 2A,B*), which resulted in the same uterine hypoplasia phenotype by PND 10 (*Figure 1C*). Finally, to confirm that the signaling in the 'inhibited progenitor' was dependent on the canonical MIS receptor, we treated *Misr2*-deficient female mice (Misr2[-/-]) with AAV9-MIS, which failed to recapitulate the uterine hypoplasia phenotype (*Figure 5—figure supplement 2B,C*). Together, these results revealed a cell-autonomous effect of MIS in the subluminal mesenchyme, intrinsic to the uterus, and dependent on *Misr2*, in which progenitors normally specified to form endometrial stromal layers at PND1-6 are inhibited by MIS, leading to long-term infertility.

## MISR2+ subluminal mesenchymal cells are transiently present in the developing embryonic human uterus

We sought to determine whether uterine subluminal mesenchymal cells expressing *MISR2* are also present in the human female fetus preceding production of the ligand (MIS) by the ovary. We used paraffin-embedded archival tissue of human female reproductive tract from fetuses ranging from 22 weeks (wk) to 37 weeks of gestation. Using the adjacent fetal ovary tissue as a positive control for MISR2 *RNAish*, we analyzed the uterus at different developmental stages, revealing a

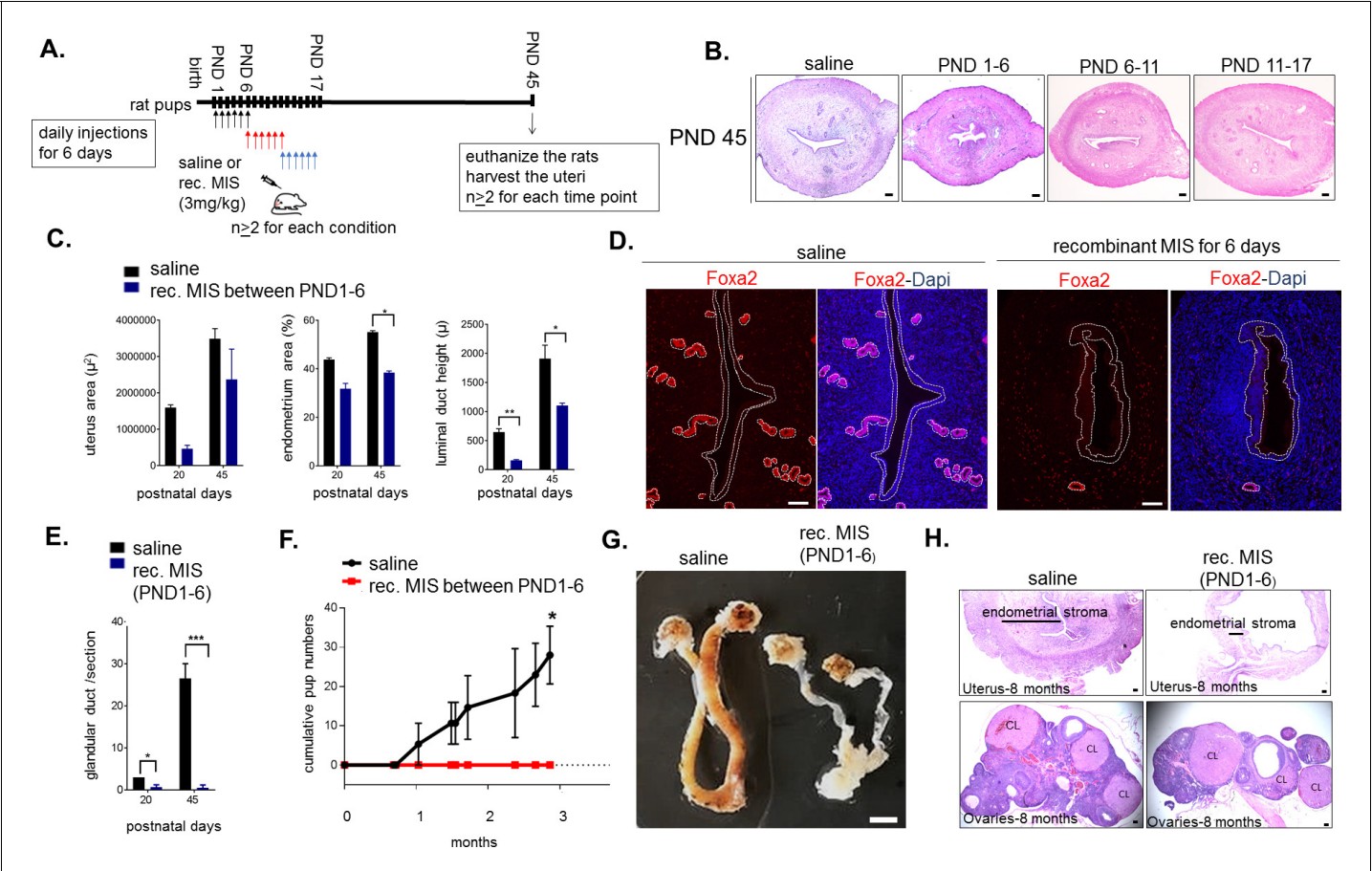

**Figure 5.** Temporary treatment with MIS during only for the first 6 days of uterine developmental is sufficient to cause complete infertility in adulthood. (A) Rat pups were injected daily with recombinant MIS protein (3 mg/kg) (MIS), or saline control (CTL), from PND1-6, PND6-11, or PND11-17. (B) Uterine morphology was analyzed on PND45 with H and E stained transverse sections (n > 2). Scale bars are 100 µm. (C) Total uterus area, the percentage of the endometrial area, and the luminal duct height in CTL and MIS-treated (from PND1-6) were compared at PND20 and 45. (D) Foxa2 immunofluorescence (red) on CTL and MIS-treated uteri (from PND1-6) was analyzed at PND20. Scale bars = 50 µm. (for C and D: for PND20, n = 2 for control, n = 3 for MIS; for PND45, n = 2 both for control and MIS, mean ± SEM, unpaired Student's t test * (p≤0.05), ** (p≤0.01)). (E) Endometrial gland counts were compared in CTL and MIS-treated uteri (from PND1-6) at PND20 and 45 from H and E sections. (F) Cumulative pups per females in 3 months continuous mating studies of the control and MIS-treated rat uteri. (n = 3 both for the control and the treated, unpaired Student's t test mean ± SEM, * (p≤0.05), *** (p≤0.001)). (G) Gross morphology of the CTL and the MIS-treated (PND1-6) uteri at 8 months of age. Scale bar = 0.5 cm. (H) Uterine transverse sections (top) and ovaries (bottom) of the control and MIS-treated rats at 8 months of age. CL stands for corpus luteum. Scale bars = 100 µm.

DOI: https://doi.org/10.7554/eLife.46349.025

The following source data and figure supplements are available for figure 5:

**Source data 1.** Data, number of replicates and p values of significance between the control and recombinant MIS-treated uterine samples for histomorphological analysis.
DOI: https://doi.org/10.7554/eLife.46349.028
**Figure supplement 1.** Neonatal exposure to MIS causes uterine hypoplasia but does not affect sex steroid levels or ovarian function.
DOI: https://doi.org/10.7554/eLife.46349.026
**Figure supplement 2.** Uterine hypoplasia in response to treatment with exogenous MIS does not require ovaries and is dependent on MISR2.
DOI: https://doi.org/10.7554/eLife.46349.027

spatiotemporal pattern of expression strikingly similar to that of the rodents (*Figure 6A–C*). *MISR2+* cells were present in the same location in the fetal uterus, directly adjacent to the lumen at 22 weeks of gestation and receded at later timepoints (24wk, 37wk), coinciding with the production of MIS by the human ovary, which is thought to begin at 24 weeks of gestation (*Kuiri-Hänninen et al., 2011*). To determine if candidate genes suspected to cause Mullerian aplasia or hypoplasia in humans (*Nik-*

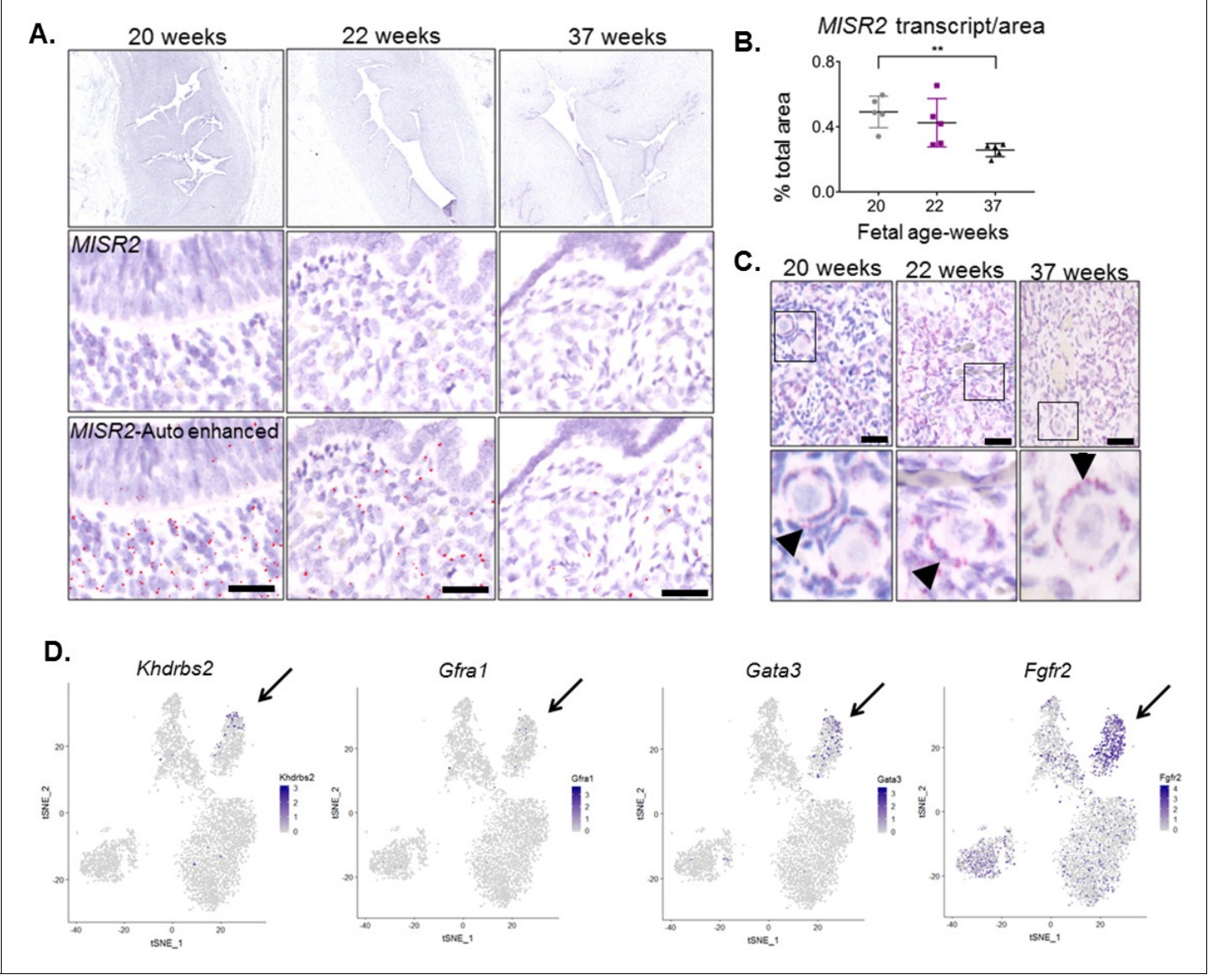

**Figure 6.** *MISR2* is expressed in the subluminal mesenchymal cells of the developing human fetus, and progenitor genes may be implicated in Mullerian aplasia in humans. (**A**) The *MISR2* expression pattern in human fetal uteri was analyzed by RNAish in fixed tissue sections. First row consists of stitched images of human fetal uteri at embryonic weeks 20, 22, and 37 (approximately 60 images (20x objective) per developmental time point were used for stitching). Middle-row panels show a higher magnification image for each subsequent time point. Red dots (*MISR2* transcripts) were quantified using the Keyence BZ-X800 analysis software. Software-enhanced red dots in the bottom-row panels show the quantified region by the automated system. Scale bars = 25 μm (**B**) MISR2 transcripts were quantified per cell area from five random sections of 20, 22 weeks, and 37 weeks fetal tissue. (unpaired Student t-test, five section per one sample, per time point, **(p≤0.01)). (**C**) Adjacent human fetal ovaries were used as internal positive controls for the *MISR2 RNAish* analysis. Black arrowheads indicate granulosa cells from primordial follicles positive for MISR2 transcripts. Scale bars = 25 μm. (**D**) Candidate genes of Mullerian aplasia in humans have enriched expression in the 'inhibited progenitor' cluster (black arrows) from the t-SNE plots of scRNAseq in rats.

DOI: https://doi.org/10.7554/eLife.46349.029

*Zainal et al., 2011*) may be present in the developing rat uterus, we analyzed their expression in our cell atlas, revealing the enrichment of several candidates within the 'inhibited progenitor' cluster (*Figure 6D*).

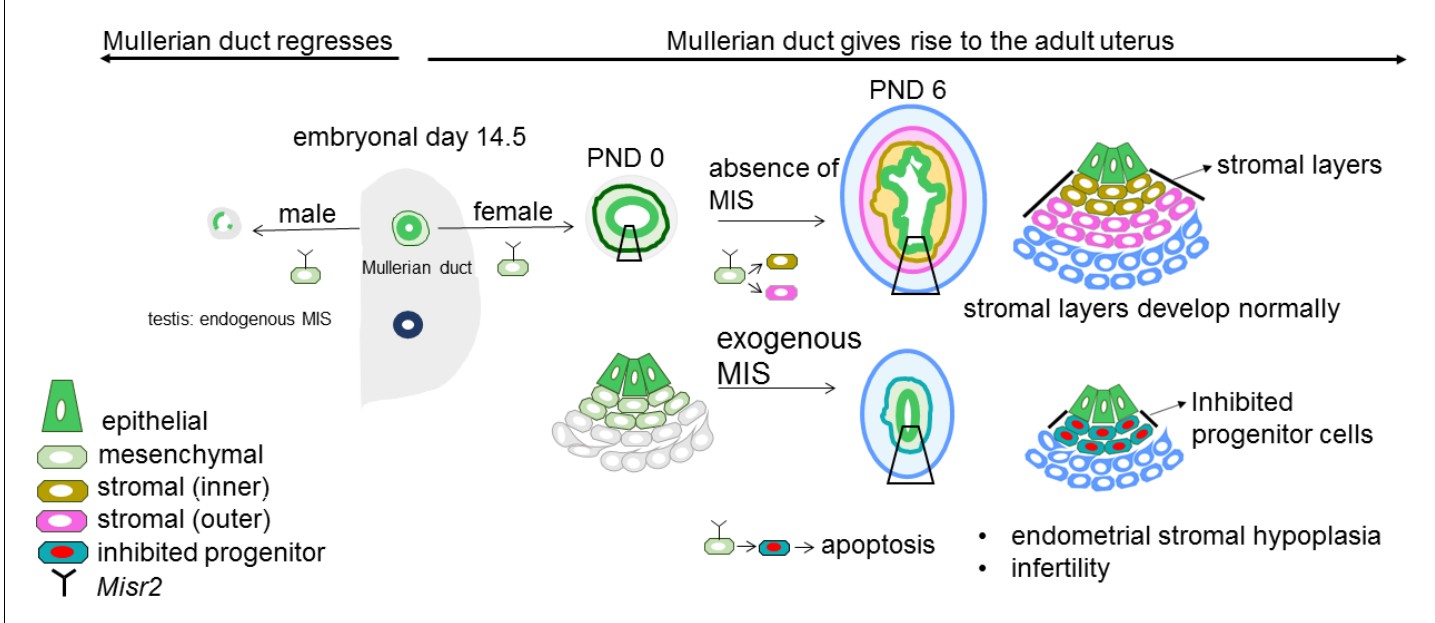

**Figure 7.** Representative schema of the *Misr2+* subluminal mesenchyme development. Mullerian inhibiting substance receptor-2 is expressed in a specific subluminal mesenchymal cell type surrounding the Mullerian duct epithelium during early fetal urogenital ridge development of both sexes. In male embryos, secretion of MIS, the ligand of Misr2, by the embryonic testes causes the Misr2+ mesenchymal cell to trigger regression of the Mullerian duct epithelium. In the female, the Misr2+ mesenchymal cells persist postnatally, and give rise to the endometrial stromal layers of the adult uterus. If females are exposed to MIS during the first week of uterus development, these subluminal progenitor cells can be reprogrammed to undergo apoptosis instead of developing into the endometrial stromal layers, resulting in uterine hypoplasia and future infertility.

DOI: https://doi.org/10.7554/eLife.46349.030

## Discussion

Single-cell RNA sequencing of the PND6 uterus revealed that the subluminal Mullerian duct mesenchyme contains a previously uncharacterized cell type (*Figure 7*) that plays a crucial role in the specification of the endometrial stroma during neonatal uterine development. We hypothesize that these *Misr2+* cells represent stromal progenitors, which normally gives rise to the inner and outer endometrial stromal layer around PND6 in mice and rats (*Figure 1B*). Surprisingly, these *Misr2+* putative stromal progenitors retain sensitivity to inhibition by MIS postnatally, and can be reprogrammed to undergo apoptosis instead of developing into the endometrial stromal layers if exposed to MIS (*Figure 7*).

Although postnatal MIS exposure is no longer able to induce regression of the uterine luminal epithelium, its normal function in male fetuses (*Jost, 1947*), it does irreversibly block its ability to form endometrial epithelial glands. We speculate that this retained sensitivity to MIS in the female may be a vestigial pathway of the male, which is normally silenced in females prior to the emergence of secretion of MIS by the ovary. However, it is unlikely that MIS itself is an important developmental trigger regulating endometrial stroma development, since the uterus develops normally in both Mis and Misr2 knockout mice (*Behringer et al., 1994*)(*Mishina et al., 1999*) (*Mishina et al., 1999*).

The postnatal response of the Misr2+ putative progenitors to MIS provides some unique insights into the developmental pathways elicited during fetal Mullerian duct regression. The nascent fetal Mullerian duct is mesoepithelial in origin, being derived from the invagination of the coelomic epithelium, and begins further differentiation into epithelium proper coincidentally with the timing of regression. Others have suggested that this epithelial differentiation may subsequently restrict the ability of the ductal cells to undergo the epithelial to mesenchymal transition characteristic of ductal regression (*Allard et al., 2000*). This raises the possibility that the *Misr2+* mesenchyme in the neonatal female may be responding to MIS similarly to male urogenital mesenchyme, but that the neonatal epithelium is unable to regress in response to those signals. Supporting this interpretation of recapitulated mesenchymal regression is the ectopic expression of many of the same genes and

pathways previously identified in the regressing male fetal Mullerian duct (*Bambi, Smad6, Wif1, etc.*) (*Mullen et al., 2018*).

Even though gain of function MIS or MISR2 mutations have not been reported in women with Mullerian anomalies, the uterine hypoplasia observed in the present study is suggestive of Mayer-Rokitansky-Küster-Hauser syndrome (MRKH), also known as Mullerian aplasia which affects 1 in 4500 women (*Nik-Zainal et al., 2011*). Genes expressed in the Misr2+ putative progenitor cells in response to MIS treatment likely represent either pathways of Mullerian duct regression or of uterine endometrial stroma progenitor development. Therefore, we speculate that the markers described herein may represent candidate genes underlying developmental disorders of the Mullerian duct. Efforts to identify causative genes within regions of copy number variation in patients affected with Mullerian aplasia have turned up candidate genes such as *KHDRBS2*, and *GFRA1* (*Nik-Zainal et al., 2011*), which we see uniquely expressed in the 'inhibited progenitor' cluster (*Figure 6D*). Similarly, both *Gata3* which causes hypoparathyroidism, sensorineural deafness, renal anomaly (HDR) syndrome with uterine hypoplasia (*Van Esch et al., 2000*) and *Fgfr2*, which causes disorders of sexual dimorphism in males (*Bagheri-Fam et al., 2015*; *Barseghyan et al., 2018*), and decidualization defects in females (*Filant et al., 2014*), are highly expressed in the inhibited MISR2+ progenitors (*Figure 6D*).

The importance of the endometrial stroma in the formation of endometrial glands and fertility has been demonstrated in multiple mouse models such as Wnt4 mutant mice, and neonatal diethylsilbesterol (DES) treatments, both of which carry phenotypes of endometrial glandular dysplasia also observed in our rodents with postnatal exposure to MIS (*Herbst et al., 1980*; *Medlock et al., 1988*; *Hayashi et al., 2011*; *Prunskaite-Hyyryläinen et al., 2016*). These mouse models display endometrial stromal hypoplasia, which precludes glandular development as a result of the disruption of Wnt pathways and the communication between stromal and epithelial compartments. Our single-cell transcriptomic analysis identified distinct inner and outer endometrial stromal layers with molecular signatures that might presage specialized stromal functions, such as regulation of the adjacent epithelium or myometrium. A comprehensive characterization of such paracrine signals across the cell types of our control and MIS-treated uterine atlases using the CellPhoneDB algorithm (*Vento-Tormo et al., 2018*) revealed the immense complexity of those cellular interactions, and their dysregulation by MIS (e.g. Wnt signaling, *Figure 4B*). MIS treatment is likely especially disruptive to the interaction between the inner endometrial stroma and adjacent luminal epithelium, as seen by the absence of expression of *Bmp7*, and coincidental downregulation of *Msx2* in those cell types, respectively, which have been previously implicated in the coordination of endometrial gland formation in the uterus (*Phippard et al., 1996*; *Kodama et al., 2010*; *Yin et al., 2015*) and decidualization defects (*Daikoku et al., 2011*; *Monsivais et al., 2017*). The nature of the paracrine signals emanating from the inner stroma regulating epithelial development, and their dysregulation during MIS treatment, including the ephrin, notch, Igf, Tnf, Mdk/Ptn, Tyro3, and Fn1 pathways (*Figure 3*, *Figure 3—figure supplement 4*), should be systematically investigated in future studies.

Furthermore, the involvement of premature exposure of the developing Mullerian ducts to MIS in disorders of sexual differentiation of the female is not fully appreciated (*Chen et al., 2014*; *Van Batavia and Kolon, 2016*). Moreover, the recent identification of an Misr2+ subepithelial adult endometrial stem cell suggests that some neonatal endometrial stromal sprogenitor may persist into adulthood, where they could play a role in endometrial homeostasis and repair (*Yin et al., 2019*). Interestingly, we have recently reported an increased incidence of preterm birth in PCOS patients with high circulating MIS (*Hsu et al., 2018*) raising the possibility of a causative link between high MIS exposure and uterine dysfunction. Conversely, the complete infertility resulting from a short treatment with MIS could have useful applications in the veterinary settings where it may be used as a permanent contraceptive that does not affect ovarian function (*Hay et al., 2018*).

Finally, given our findings that MIS is a potent inhibitor of MISR2+ putative endometrial stromal progenitors during uterine development, and that this cell type is likely also active in fetal human uteri, it would be of interest to explore possible clinical applications of MIS, or related pathways, in the context of Mullerian development pathologies, uterine infertilities, and endometrial stromal cancers, should these pathways become reactivated in those tumors (*Chiang and Oliva, 2013*).

# Materials and methods

## Key resources table

| Reagent type (species) or resource | Designation | Source or reference | Identifiers | Additional information |
|---|---|---|---|---|
| Genetic reagent (*M. muculus*) | B6;129S7-Amhr2tm3 (cre)Bhr/Mmnc | PMID: 12368913 | RRID_ MGI:3042214 | Dr. Richard R Behringer, MD Anderson Cancer Center |
| Genetic reagent (*M. muculus*) | C57BL/6-Tg(UBC-GFP) 30Scha/J | Jackson Laboratory | stock #004353 | Hongkui Zeng, Allen Institute for Brain Science |
| Genetic reagent (*M. muculus*) | FVB/NCrl | Charles River | #207 | |
| Genetic reagent (Rat) | Sprague Dawley | Envigo | | |
| Peptide, recombinant protein | LR-MIS | (*Pépin et al., 2013*) | | |
| Recombinant DNA reagent | AAV9-LRMIS | (*Pépin et al., 2015*) | | |
| Antibody | Smooth muscle alpha action (SMA) (Rabbit polyclonal) | Abcam | #5694 | (1:300), IF |
| Antibody | Vimentin (rabbit monoclonal) | Abcam | #92547 | (1:300), IF |
| Antibody | Foxa2 (rabbit polyclonal) | LifeSpan Biosciences | #138006 | (1:500), IF |
| Antibody | Cleaved caspase-3 (rabbit polyclonal) | Cell signaling | #9661S | (1:200), IF; (1:500) IHC |
| Antibody | E-cadherin (Cdh1) (rat monoclonal) | Invitrogen | #13–1900 | (1:200), IF |
| Antibody | Alexa flour 488 donkey anti rat IgG | Invitrogen | #A21208 | (1:500) |
| Antibody | Alexa flour 555 donkey anti rabbit IgG | Invitrogen | #A31572 | (1:500) |
| Antibody | Alexa flour 568 anti rabbit IgG | Invitrogen | #A10042 | (1:500) |
| Commercial assay or kit | RNA scope 2.5 HD red detection kit | ACD bio | #322360 | |
| Commercial assay or kit | the target retrieval and protease plus reagents | ACD bio | #322330 | |
| Commercial assay or kit | MIS commercial ELISA | Beckmen | #A73818 | |
| Commercial assay or kit | REDExtract-N-Amp Tissue PCR Kit | Sigma | #SLBT8193 | |
| Other | Bambi (*M. muculus*) (NM_026505.2) | ACD bio | #523071 | commercial probe |
| Other | Bmp7 (*M. muculus*) (NM_007557.3) | ACD bio | #407901 | commercial probe |
| Other | CD24a (*M. muculus*) (NM_009846.2) | ACD bio | # 432691 | commercial probe |
| Other | Cpxm2 (*M. muculus*) (NM_018867.5) | ACD bio | # 559759 | commercial probe |

*Continued on next page*

*Continued*

| Reagent type (species) or resource | Designation | Source or reference | Identifiers | Additional information |
|---|---|---|---|---|
| Other | Enpp2 (*M. muculus*) (NM_001136077.1) | ACD bio | # 402441 | commercial probe |
| Other | Hdac4 (*M. muculus*) (NM_207225.1) | ACD bio | # 416591 | commercial probe |
| Other | Misr2 (Amhr2) (*M. muculus*) (NM_144547.2) | ACD bio | # 489821 | commercial probe |
| Other | Misr2 (Amhr2) (*Rat*) (NM_030998.1) | ACD bio | # 517791 | commercial probe |
| Other | Misr2 (Amhr2) (*Human*) (NM_020547.2) | ACD bio | # 490241 | commercial probe |
| Other | Msx2 (*M. muculus*) (NM_013601.2) | ACD bio | # 421851 | commercial probe |
| Other | Myh 11 (*M. muculus*) (NM_001161775.1) | ACD bio | # 316101 | commercial probe |
| Other | Smad6 (*M. muculus* and *Rat*) (NM_001109002.2) | ACD bio | # 517781 | commercial probe |
| Other | Thbs2 (*M. muculus*) (NM_011581.3) | ACD bio | # 492681 | commercial probe |
| Other | Wfikkn2 (*M. muculus*) | ACD bio | # 531321 | commercial probe |
| Software, algoritm | R | R Project for Statistical Computing | https://scicrunch.org/resolver/SCR_001905 | |
| Software, algoritm | BZ-X800 analysis software | Keyence | https://www.keyence.com/landing/microscope/lp_fluorescence.jsp | |
| Software, algoritm | GraphPad Prism, version 7 | Graphpah | | |

## Animals

This study was performed in accordance with experimental protocols 2009N000033 and 2014N000275 approved by the Massachusetts General Hospital Institutional Animal Care and Use Committee. Strains of Sprague–Dawley (purchased from Envingo) and Friend leukemia virus B (FVB) (purchased from Charles River Laboratories) were used for rat and mouse experiments, respectively. Misr2/Amhr2-Cre knock-in mice were purchased from the Mutant Mouse Regional Resource Centers (MMRRC) (strain B6;129S7-Amhr2tm3(cre)Bhr/Mmnc, backcrossed with C57BL/6J) (*Jamin et al., 2002*). Tail genotyping of the Misr2-cre knock-in and WT mice were done with REDExtract-N-Amp Tissue PCR Kit (Sigma, #SLBT8193) with the previously described sets of primers (*Jamin et al., 2002*).

## AAV9-MIS and recombinant protein treatment

The adeno-associated virus serotype 9 (AAV9) gene therapy vector was used for sustained delivery of a higher concentration of human MIS analog (LR-MIS) as described (*Pépin et al., 2015*). To test the effect of continuous MIS exposure on uteri, rats or mice were injected subcutaneously with AAV9-recombinant human LR or RF MIS (AAV9-MIS) on postnatal day 1, and their uteri were harvested at different time points for histological analysis (5E10 particles/pup). Blood was collected by cardiac puncture at endpoint, and centrifuged at $900 \times g$ for 10 min at room temperature from control and AAV9-MIS-treated female rat pups on PND3, 5, 6, 10, 15, and 30 (n > 2). To validate the RNA scope markers on tissue sections, mice were injected with AAV9-MIS on PND 1 (1E10 particles/

pup), and sacrificed on PND 6 (n = 3 both for the control and the treated). For each time point, the uteri were cut radially in halves. One half was fixed in formalin for histological analysis and immunohistochemistry, the other half was flash-frozen for RNA isolation and qPCR analyses. In mice, the paraffin-embedded fixed tissue was sectioned for RNAish analysis, while in rats it was used for histomorphological analyses.

To test the window of sensitivity to exogenous MIS during uterus development, rats were injected subcutaneously with a human recombinant MIS (LR-MIS) protein (*Pépin et al., 2013*) daily (3 mg/kg/day) for 6 days starting from PND1-6, PND6-11, or PND11-17 (*Figure 5A*). Uteri of rat pups injected from PND1-6 were harvested and fixed on 20, and 45 (for PND20, n = 2 for control, n = 3 for treated, for PND45 n = 2 both for the control and the treated). Uteri of the rat pups injected from PND6-11, or PND11-17, were harvested and fixed on PND45 (n > 2 for all) (*Figure 5A*).

## Gonadectomy

To rule out the involvement of ovarian hormones as a contributor to the MIS-induced uterine hypoplasia, rat pups were gonadectomized two days after birth (n = 4), prior to receiving MIS treatment (*Figure 5—figure supplement 2A–B*). For gonadectomy, the rat pups were anesthetized using isofluorane. Bilateral longitudinal incisions were made in the mid dorsal line through the skin and musculature one-third the distance between the base of the tail and the neck, directly over the position of the ovary. The ovaries together with the oviducts were extruded through the incision via the ovarian fat pad. The ovarian vasculature was ligated between the oviduct and uterine horn and the ovaries and oviducts were resected. The muscle layer was closed with silk sutures, and the outer skin was closed with a single metal clip. The rats were then kept warm until they had completely recovered from the anesthesia. Analgesia was provided for 3 days with Carprofen PO. Two of the rat pups were treated with AAV9-MIS, while the controls were treated with empty vector (5E10 particles, subcutaneously, n = 2 both for control and treated) 6 hr after the surgery. The pups were then euthanized on PND10, and their uteri were fixed for histomorphological analysis (*Figure 5—figure supplement 2A–B*).

## Misr2$^{cre/cre}$ transgenic mice

To verify that the MIS Receptor 2 (Misr2) is the mediator of the uterine phenotype caused by exogenous MIS, *Misr2$^{cre/+}$* and *Misr2$^{cre/cre}$* transgenic mice (*Jamin et al., 2002*) were treated with empty vector or AAV9-MIS (5E10 particles, subcutaneously) on day 1 and their uteri were analyzed on PND20 (*Figure 5—figure supplement 2C–D*) (n = 2 for *Misr2$^{cre/+}$*, n = 1 for *Misr2$^{cre/cre}$*). *Misr2$^{cre/cre}$* males had retained Mullerian ducts confirming the loss-of-function of the Misr2 (*Jamin et al., 2002*).

## ELISA

The Beckman AMH ELISA (Beckman, #A73818), which can detect both endogenous murine MIS and exogenous human MIS secreted by the AAV9-MIS infected muscles was used to measure the serum MIS levels. To detect the murine endogenous MIS levels during the developmental time span of rat females, serum from control PND1, 4, 6, and 20 rats, as well as AAV9-MIS-treated rats on PND six were measured by ELISA (n = 3 for PND4, 6 and 20, and n = 2 for PND 1, n = 3 for AAV9-MIS treated rat).

Murine Estradiol and Testosterone serum levels were measured with specific ELISAs at the Ligand Assay and Analysis Core of the Center for Research in Reproduction at University of Virginia School of Medicine under a cooperative agreement (The core is supported by the Eunice Kennedy Shriver NICHD/NIH (NCTRI), Grant P50-HD28934). (n = 3 for PND3 and 30 control and treated, and PND 5 control; n = 2 for PND five treated; PND6, 10, 15 control and treated animals).

## Fertility tests

Sprague–Dawley rats were injected subcutaneously with 3 mg/kg/day of recombinant MIS (LR-MIS) or 20 µl saline (vehicle control) from PND1-6. One MIS-treated and one sibling control female were caged with one experienced breeder male (n = 3 cages) at 6 weeks of age. The male was separated from the cage after pregnancy was identified and returned after the pups were weaned. The total number of pups and litters from each female was monitored for a period of 4 months.

## Histology, immunofluorescence (IF), immunohistochemistry (IHC) and RNA in situ hybridization

Dissected uteri and ovaries were fixed in 4% (wt/vol) paraformaldehyde at 4°C (for histology and immunofluorescence) or in 10% neutral buffered formalin at room temperature overnight (for RNA-ish). Tissues were embedded in paraffin blocks in an automated tissue processor (Leica #TP1020). 5 µm transverse uterine sections from the middle of the uterine horn (i.e 'b' in *Figure 1—figure supplement 1D*) were used for hematoxylin and eosin (H&E) staining, immunofluorescence (IF), and RNAish using the RNA scope (ACD bio) system. Archival human fetal tissue sections were provided by the Massachusetts General Hospital, Gynecological Pathology Department through an IRB approved protocol (IRB 2007P001918).

Tissue sections were rehydrated for IF in an alcohol series after deparaffinization in xylene. Antigen retrieval was performed by parboiling in 10 mM sodium citrate (pH 6.0), cooling at room temperature, blocking in 3% bovine serum albumin (BSA) in Tris-buffered solution (TBS) for 1 hr, followed by three washes (10 min each) in TBS and the sections incubated in primary antibody overnight at 4°C. For double-labeling, the slides were blocked after washes, and then incubated with a second primary antibody overnight at 4°C. The sections were then incubated in fluorescently conjugated secondary antibodies (Alexa Fluor 555-conjugated donkey anti-rabbit IgG antibody, # A31572; Alexa Fluor 488-conjugated donkey anti-rabbit IgG, #A21206) for one hour at room temperature and cover-slipped with vectashield mounting medium with DAPI (Vector Laboratories # NC9265087). For immunohistochemistry (IHC), Dako EnVision + System horseradish peroxidase (HRP) Labeled Polymer Anti-Rabbit was used as the secondary antibody (#K4002), and the HRP signal was detected using the DAKO detection system (Dako, #K5007). Antibody dilutions for IF and IHC were as follows: Smooth muscle alpha action (SMA) (1:300, abcam, #5694), Vimentin (1:300, abcam, #32547), Foxa2 (1:500, LifeSpan Biosciences, #138006, 1:500), cleaved caspase-3 (1:50, cell signaling, #9661S), E-cadherin (Cdh1) (1:200, Invitrogen #13–1900).

RNAish was performed with the manual RNAscope 2.5 HD Reagent Kit (RED) (ACD Bio, # 322350) following the manufacturer's instructions as previously described (*Wang et al., 2012*). The tissue sections were hybridized with pre-designed or custom-designed probes spanning mRNAs of the target genes (see Table S1 for accession number, target region, and catalog number of each gene) in the HybEZ hybridization oven (ACD Bio) for 2 hr at 40°C, following deparaffinization in xylene, dehydration, peroxidase blocking, and heat-induced epitope retrieval by the target retrieval and protease plus reagents (ACD bio, #322330). The slides were then processed for standard signal amplification steps, and a red chromogen development was performed using the RNAscope 2.5 HD (Red) detection Kit (ACD Bio, #322360). The slides were then counterstained in 50% hematoxylin (Dako, #S2302) for 2 min, air-dried and coverslipped with EcoMount.

## Histomorphological analysis and quantification of RNA-scope images

Middle sections ('b' in the scheme of *Figure 1—figure supplement 1D*) of control and MIS-treated rat uteri were stained with H and Es for histomorphological analyses, which were conducted at different developmental time points) (*Figure 2—source data 1*, *Figure 5—source data 1*). Luminal duct height, area of the whole uterus, and area of the endometrium were calculated from the transverse sections using the image J software. For *Figure 2*, n = 3 for PND3 control and treated; n = 2 for PND6, 20 control and treated, n = 1 for PND10 control and treated samples. For *Figure 4*, n = 2 for the PND20 control; PND45 control and the treated; n = 3 for the treated PND20 sample (*Figure 2—source data 1*).

Human fetal tissue sections were imaged by the Keyence BZ-X800 microscope at 20x resolution, and the RNAish stains were auto-quantified by BZ-X800 analysis software. Approximately 60 images were obtained per stained section, and stitched together for the top panel of *Figure 6a*. 5 random 20x images were selected per time point for analysis, and areas of red RNA scope dots (Misr2 transcripts amplified by RNA-scope) were detected and labeled based on hue (see red dots on the bottom section of *Figure 4a*, labeled as 'enhanced Misr2'). Total cell area was calculated by setting the masked area as the hematoxylin-stained region. RNAish dots/total area were auto-calculated by the same settings in 5 random 20X sections of human fetal tissues from 20, 22, and 37 weeks of gestation, n = 1 for each time point.

## Quantitative PCRs

Total RNA was extracted from the uteri of control and AAV9-MIS treated rats at different time points (*Figure 3—source data 3*) using the Qiagen RNA extraction kit. For all the samples, cDNA was synthesized from 500 ng total uterine RNA using SuperScript III First-Strand Synthesis System for RT-PCR according to manufacturer's instructions using random hexamers (Invitrogen, # 18080–051). The primers were designed to span the exon-exon junctions of the target genes (see *Figure 3—source data 3* for complete list of primers) to avoid genomic DNA contamination. Expression levels relative to 18S (for *Acta2*) and *Gapdh* (for all the other genes analyzed) were calculated by using cycle threshold (Ct) values logarithmically transformed using the $2^{-\Delta Ct}$ function and the average value of the relative expression levels were normalized to PND three control set; and fold changes were calculated relative to PND three control time point with three technical replicates per sample. For *Acta2*, *Tgln*, and *Foxa2* expression levels were normalized to PND six control set. Sample sizes and p values for each gene and time point are listed in *Figure 3—source data 3*.

## Generation of single cell (sc) suspension for scRNAseq

Newborn rats (PND1) were injected with 5E10 particles of AAV9-MIS (N = 3) or AAV9 empty particle controls (N = 3). On PND6, rat pups were sacrificed and the uterine tissue was microdissected, taking care to exclude the oviduct, cervix, and ureter. Both uterine horns from each animals (N = 3 per group) were combined and placed in 5 mL of dissociation medium (82 mM $Na_2SO_4$, 30 mM $K_2SO_4$, 10 mM Glucose, 10 mM HEPES, and 5 mM $MgCL_2$ - $6H_2O$, pH 7.4) containing 15 mg of Protease 23 (Worthington), 100 U Papain with 5 mM L-Cysteine and 2.5 mM EDTA (Worthington), and 1333 U of DNase 1 (Worthington) prewarmed to 34C. The samples were placed on a rocker at 34C for 15 min. The medium was then removed with a pipette and replaced with 5 mL chilled (4C) stop medium (dissociation medium containing 0.025% BSA) supplemented with 0.5 mg trypsin Inhibitor and 0.5 mg ovomucoid protease Inhibitor. Samples were triturated 10 times with a 5 mL pipette, then 10 times with a 1000 μl micropipette, and finally filtered through a prewetted 100 μm filter. Filtered samples were spun at 1000 g for 10 min, and the cell pellet was resuspended in 1 ml of stop solution with 267 U of DNase. This step was repeated twice. Ten microliters was removed from each sample and combined with trypan blue to assess viability and concentration of cells. The cell mixture was spun a final time and resuspended in stop solution containing 20% Optiprep (Sigma) to a concentration of 150,000 cells/mL for inDrop sorting.

Single-cell RNA sequencing (inDrop) inDrop microfluidic sorting was performed as previously described (*Hrvatin et al., 2018*) generating two libraries of approximately 5000 cells from each combined (N = 3 rats) cell suspensions (control and MIS). Transcripts were processed as previously described and samples were sequenced on a NextSeq 500 (Ilumina) in a single combined lane (*Hrvatin et al., 2018*).

## Data analysis of scRNAseq

The analysis of the demultiplexed data was performed using the Seurat package in 'R' (*Butler et al., 2018*). Filtering parameters were set to remove cells with fewer than 200 genes, and those with more than 3000 genes and/or 10,000 UMIs from the dataset. Dataset included 6811 control cells, and 2990 MIS treated cells which were jointly normalized. The Pearson correlation coefficient of UMI and nGene across the dataset was 0.97. Principal component analysis was performed using 20 dimensions and FindClusters parameters were set at k.param = 30, k.scale = 25, and prune. SNN = 0.0667 with a resolution of 0.6 using 5346 variable genes. Data set and R codes are presented in *Figure 3—source data 1*. For differential expression analysis of the myometrium and the epithelial cell clusters, data sets are presented in *Figure 3—source data 2* and *Figure 4—source data 2*.

## CellPhoneDB analysis

CellPhoneDB predicts signaling between cell clusters through analyzing co-expression of known Human receptors and secreted proteins. We downloaded ortholog information for Rat (Rnor_6.0) and Human (GRCh38.p12) from ENSEMBL (Release 95). For non one-to-one orthologs, we assigned the maximum of gene expression values for all Rat to Human mappings. CellPhoneDB was

subsequently run using default parameters (*Vento-Tormo et al., 2018*). The resulting data set is presented in *Figure 4—source data 1*.

## Statistical analysis

For serum MIS ELISA measurements of the control rats, two-way ANOVA analysis was used. For the serum ELISA measurements of the control and the treated rats on PND6, the histomorphology, and the staining analyses, unpaired Student's t test was used to compare the control and the treated samples using the Prism software (Graphpad version 8.0). p values are presented in *Figure 2— source data 1*, and *Figure 3—source data 3*, and *Figure 5—source data 1*.

## Acknowledgements

We thank Bernardo Sabatini for his constructive feedback and assistance with the scRNAseq experiments. We acknowledge Abigail Alexander, Raghav Mohan, Selena Yuan, in performing histological stains.We also thank Nobuhiro Takahashi, Yi Li, Caroline Coletti, Nicole Foxworth, Ryo Hotta, and Sarah Mustafa Eisa for their technical help. This study was funded in part by Huiying Fellowship (HDS), Grant MG14-S06R from the Michelson Found Animal Foundation (to DP, PKD, and GG); a Sudna Gar Fellowship (DP); the Massachusetts General Hospital Executive Committee on Research (ECOR) (DP and PKD); and royalties from the use of the MIS ELISA in infertility clinics around the world.

## Additional information

### Funding

| Funder | Grant reference number | Author |
|---|---|---|
| Michelson Prize and Grants | MG14-S06R | Patricia K Donahoe<br>David Pépin<br>Guangping Gao |
| Huiying Fellowship | | Hatice Duygu Saatcioglu |
| Sudna Gar Fellowship | | David Pépin |
| Massachusetts General Hospital | Massachusetts General Hospital Executive Committee on Research (ECOR) | Patricia K Donahoe<br>David Pépin |

The funders had no role in study design, data collection and interpretation, or the decision to submit the work for publication.

### Author contributions

Hatice Duygu Saatcioglu, Conceptualization, Data curation, Formal analysis, Supervision, Validation, Investigation, Visualization, Methodology, Writing—original draft, Writing—review and editing; Motohiro Kano, Data curation, Formal analysis, Validation, Investigation, Methodology; Heiko Horn, Data curation, Software, Formal analysis, Validation, Writing—original draft, Writing—review and editing; Lihua Zhang, Data curation, Validation, Methodology; Wesley Samore, Resources, Data curation, Formal analysis, Investigation; Nicholas Nagykery, Resources, Investigation, Methodology; Marie-Charlotte Meinsohn, Formal analysis, Validation, Investigation, Writing—review and editing; Minsuk Hyun, Resources, Formal analysis, Methodology; Rana Suliman, Validation, Investigation; Joy Poulo, Data curation, Software, Formal analysis, Validation, Methodology; Jennifer Hsu, Formal analysis, Investigation, Methodology; Caitlin Sacha, Data curation, Formal analysis, Methodology, Writing—review and editing; Dan Wang, Conceptualization, Resources, Methodology; Guangping Gao, Conceptualization, Resources, Supervision; Kasper Lage, Resources, Software, Supervision, Project administration; Esther Oliva, Conceptualization, Data curation, Supervision, Investigation; Mary E Morris Sabatini, Conceptualization, Investigation, Methodology; Patricia K Donahoe, Conceptualization, Supervision, Funding acquisition, Project administration, Writing—review and editing; David Pépin, Conceptualization, Data curation, Formal analysis, Supervision, Funding

acquisition, Validation, Investigation, Visualization, Writing—original draft, Project administration, Writing—review and editing

### Author ORCIDs

Hatice Duygu Saatcioglu (ID) https://orcid.org/0000-0003-2210-0005
Motohiro Kano (ID) https://orcid.org/0000-0002-5443-0175
Heiko Horn (ID) https://orcid.org/0000-0003-4898-0557
Jennifer Hsu (ID) https://orcid.org/0000-0001-6928-2585
David Pépin (ID) https://orcid.org/0000-0003-2046-6708

### Ethics

Human subjects: Human fetal tissue sections were procured by the Massachusetts General Hospital, Gynecological Pathology Department through The Institutional Review Board (IRB) approved protocol (#IRB 2007P001918).

Animal experimentation: This study was performed in accordance with experimental protocols 2009N000033 and 2014N000275 approved by the Massachusetts General Hospital Institutional Animal Care and Use Committee.

### Decision letter and Author response

Decision letter https://doi.org/10.7554/eLife.46349.035
Author response https://doi.org/10.7554/eLife.46349.036

## Additional files

### Supplementary files

• Transparent reporting form
DOI: https://doi.org/10.7554/eLife.46349.031

### Data availability

Sequencing data have been deposited in OSF platform, the link is as follows: https://osf.io/27hej/.

The following dataset was generated:

| Author(s) | Year | Dataset title | Dataset URL | Database and Identifier |
| --- | --- | --- | --- | --- |
| Saatcioglu HD, Kano M, Horn H, Joy MP, Kasper L, Morris Sabatini ME, Donahoe PK, Pépin D | 2019 | Single-cell sequencing of neonatal uterus reveals an endometrial stromal progenitor indispensable for female fertility | https://osf.io/27hej/ | Open Science Framework, 27hej |

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
