## [Decision Letter]

Thank you for submitting your article "Single-cell sequencing of neonatal uterus reveals an endometrial progenitor indispensable for fertility" for consideration by *eLife*. Your article has been reviewed by three peer reviewers, and the evaluation has been overseen by a Reviewing Editor and Marianne Bronner as the Senior Editor. The following individuals involved in review of your submission have agreed to reveal their identity: Yuji Mishina (Reviewer #2).

The reviewers have discussed the reviews with one another and the Reviewing Editor has drafted this decision to help you prepare a revised submission.

Summary:

The authors seek to elucidate the cell types and pathways that are involved in early postnatal specification of the uterine compartments. This is a carefully performed expression analysis of Misr2+ mesenchymal progenitors cells during post-natal development. Mullerian ducts (MD) are the embryonic precursor of the female reproductive system. In males, these ducts regress due to the secretion of Mullerian Inhibiting Substance (MIS) from Sertoli cells of the testis. However, in females, due to the absence of MIS, these ducts persist and form oviducts, uterus, cervix and a part of the vagina. MD of both male and female's express receptors for MIS. In this study, authors have shown experimental evidence for these cells to be the progenitor cells for endometrial stroma, by exogenous MIS Rx to postnatal female mice and complimentary single-cell RNA profiling. The authors have identified much needed new markers of uterine mesenchymal cells, which would be immensely useful for understanding the pathogenesis of many female reproductive disorders. Overall, this is very a significant study addressing an important area of female reproductive biology.

Essential revisions:

Most of the comments can be addressed by additional explanations – please see below.

1) The authors interpret the Misr2-CRE lineage tracing experiment to mean that any tomato positive cells seen after birth were derived from the Misr2 progenitors observed during embryonic development, but isn't it possible that stromal cell types later turn on Misr2 in distinct cell populations that are not derived from those original progenitors? An alternative view could be that the Misr2 has promiscuous expression and thus turns on Cre in multiple cell types? Please discuss caveats to the interpretation put forth in the manuscript.

2) The single cell experiments are useful, especially in the context of WT mice. However, I am not entirely sure if the overexpression experiments are as useful. Presumably, providing ectopic MIS via AAV9 is used to investigate the natural course of MIS signaling through Misr2+ cells, but I wonder how this overexpression experiment might differ from what happens during normal sexual development. Could the doses be producing artificial effects?

3) The authors are quite liberal in using the term progenitor or the multipotent progenitor (subsection “MIS blocks the expansion and differentiation of an *Misr2*+ stromal progenitor”) cells across the manuscript. However, definitive evidence showing that these MISr2+ cells are progenitor cell population is not provided. To use the term stem/progenitor, the author should temporary label these cells and then track their fate to show these labeled cells give rise to other cell types of the uterus (multipotent) or stromal cells (unipotent progenitors). The experiment with Misr2-cre-tdtomato mice somewhat addresses this issue. However, MISr2 expression is present in the MD stroma from 13.5dpc in mice, and it is not clear whether these cells were from 13.5dpc, when Cre was activated, or day 2-6 postnatal uteri. This could be easily addressed by developing *MISr2*cre;rttaflox;teo-tdtomato mice where cells can be labeled postnatally and tracked over time. Alternatively, the authors can downplay the emphasis on stem/progenitor cells specially multipotency of these cells.

4) The authors need to explain discrepancies in their in-situ data of the MISr2 gene and MISr2-lacZ mice data where real-time lacZ expression is maintained till adulthood in females (Arango et al., 2008)

5) There is no information on how many tissue sections per mice were examined and their relative location (close to the ovary or cervix etc.). How many were used for in-situ? What depth? There are significant differences in histology and the lumen width across the length of uterus.

*Reviewer #1:*

The authors seek to elucidate the cell types and pathways that are involved in early postnatal specification of the uterine compartments. This is a carefully performed expression analysis of Misr2+ mesenchymal progenitors cells during post-natal development. MIS exposure led to impaired uterine development, implicating the developmental trajectories of Misr2 progenitors. Elegant single-cell RNA sequencing analysis was used to show that Misr2+ progenitor cells give rise to the endometrial stroma in rodents. Ectopic overexpression was delivered to post-natal uteri and this blocks the proper specification of the endometrial stroma.

The authors interpret the Misr2-CRE lineage tracing experiment to mean that any tomato positive cells seen after birth were derived from the *Misr2* progenitors observed during embryonic development, but isn't it possible that stromal cell types later turn on Misr2 in distinct cell populations that are not derived from those original progenitors? An alternative view could be that the *Misr2* has promiscuous expression and thus turns on Cre in multiple cell types? Please discuss caveats to the interpretation put forth in the manuscript.

The single cell experiments are useful, especially in the context of WT mice. However, I am not entirely sure if the overexpression experiments are as useful. Presumably, providing ectopic MIS via AAV9 is used to investigate the natural course of MIS signaling through Misr2+ cells, but I wonder how this overexpression experiment might differ from what happens during normal sexual development. Could the doses be producing artificial effects?

*Reviewer #2:*

This paper explores a previously uncharacterized cell type in uterine mesenchyme that plays a critical role for postnatal development of endometrial stroma. Mullerian duct Inhibitory Substance (MIS), also known as Anti Mullerian Hormone (AMH), is a member of TGF-β superfamily to regress the Mullerian ducts during male embryogenesis, which give rise reproductive tracts in female embryos such as the uterus and the oviduct. The authors focus on the fact that development of newborn stage uterus is compromised by dosing of MIS and identify subpopulation of stromal cells that are expressing MIS type 2 receptor (Misr2) is critical to support enlargement of uterine wall. Since MIS interacts this population to inhibit the expansion of the endometrial stroma, the authors name this population of cells as inhibited progenitors. This is an interesting and innovative piece of work since function of MIS on male embryos have been well studied while its function in females is poorly understood. There are a couple of papers reporting about MIS function in postnatal ovaries, but almost no attempts have been made to understand how MIS regulates postnatal development of uterus, despite the well-known fact that females start to produce MIS postnatally. The authors provide ample pieces of data to support their conclusions using three different models such as newborn stages of mice and rats, and human embryos. Those findings may provide molecular insight to understand pathogenesis of uterine tumors, Mullerian aplasia, and miscarriage.

*Reviewer #3:*

Mullerian ducts (MD) are the embryonic precursor of the female reproductive system. In males, these ducts regress due to the secretion of Mullerian Inhibiting Substance (MIS) from Sertoli cells of the testis. However, in females, due to the absence of MIS, these ducts persist and form oviducts, uterus, cervix and a part of the vagina. MD of both male and female's express receptors for MIS. In this study, authors have shown experimental evidence for these cells to be the progenitor cells for endometrial stroma, by exogenous MIS Rx to postnatal female mice and complimentary single-cell RNA profiling. These authors have identified much needed new markers of uterine mesenchymal cells, which would be immensely useful for understanding the pathogenesis of many female reproductive disorders. Overall, this is very a significant study addressing an important area of female reproductive biology.

Major concerns:

1) The authors are quite liberal in using the term progenitor or the multipotent progenitor (subsection “MIS blocks the expansion and differentiation of an *Misr2*+ stromal progenitor”) cells across the manuscript. However, definitive evidence showing that these MISr2+ cells are progenitor cell population is not provided. To use the term stem/progenitor, the author should temporary label these cells and then track their fate to show these labeled cells give rise to other cell types of the uterus (multipotent) or stromal cells (unipotent progenitors). The experiment with Misr2-cre-tdtomato mice somewhat addresses this issue. However, MISr2 expression is present in the MD stroma from 13.5dpc in mice, and it is not clear whether these cells were from 13.5dpc, when Cre was activated, or day 2-6 postnatal uteri. This could be easily addressed by developing MISr2cre;rttaflox;teo-tdtomato mice where cells can be labeled postnatally and tracked over time. Alternatively, the authors can downplay the emphasis on stem/progenitor cells specially multipotency of these cells.

2) The author needs to explain discrepancies in their in-situ data of the MISr2 gene and MISr2-lacZ mice data where real-time lacZ expression is maintained till adulthood in females (Arango et al., 2008)

3) There is no information on how many tissue sections per mice were examined and their relative location (close to the ovary or cervix etc.). How many were used for in-situ? What depth? There are significant differences in histology and the lumen width across the length of uterus.

---

## [Author Response]

Essential revisions:Most of the comments can be addressed by additional explanations – please see below.1) The authors interpret the Misr2-CRE lineage tracing experiment to mean that any tomato positive cells seen after birth were derived from the Misr2 progenitors observed during embryonic development, but isn't it possible that stromal cell types later turn on Misr2 in distinct cell populations that are not derived from those original progenitors? An alternative view could be that the Misr2 has promiscuous expression and thus turns on Cre in multiple cell types? Please discuss caveats to the interpretation put forth in the manuscript.

We agree with the non-mutually exclusive interpretations of the lineage tracing experiment put forth by the reviewer. The misr2-cre reporter will mark both the descendants of the well-established MISR2+ mesenchyme surrounding the Mullerian duct where it is first expressed at E13 (Arango et al., 2008), as well as any other cell type which may subsequently express MISR2 after E13. While the lineage tracing experiment (Figure 1—figure supplement 1A) demonstrates that the endometrial stroma, but not its epithelium, are derived from Misr2+ cells, it is not in itself a definitive proof of the direct lineage between subluminal cells and endometrial stroma, although it strongly supports this hypothesis.However, the focus of this study regards the postnatal development of these subluminal Misr2+ cells, which we show can be inhibited by MIS, resulting in the absence of endometrial stroma formation, which is also consistent with a progenitor interpretation. We have carefully highlighted the tentative nature of this interpretation by referring to these cells as “putative” endometrial stromal progenitors. Furthermore, these interpretations and their caveats have been highlighted in the manuscript (Results section paragraph one), and in Figure 1B with a schematic model to clarify the focus on the cell type of interest (Misr2 + subluminal cells).

2) The single cell experiments are useful, especially in the context of WT mice. However, I am not entirely sure if the overexpression experiments are as useful. Presumably, providing ectopic MIS via AAV9 is used to investigate the natural course of MIS signaling through Misr2+ cells, but I wonder how this overexpression experiment might differ from what happens during normal sexual development. Could the doses be producing artificial effects?

We thank the reviewer for these comments. In this study, the ectopic expression of MIS in the female was used as a tool to induce an artificial postnatal inhibition of the MISR2+ subluminal mesenchyme, at a time when ovaries do not yet produce MIS (Figure 1—figure supplement 1B and C). This allowed us to explore the development of this cell type by preventing their expansion and differentiation, thus revealing their ultimate role in endometrial stromal formation. We interpret the continued Misr2 expression in females beyond the embryonic sex determination period of the ducts as a vestigial pathway of the male. The observation that Misr2expression in this subluminal mesenchyme declines prior to MIS production by the ovary supports the hypothesis that MIS itself likely does not play an important role in the normal development of the endometrial stroma. This interpretation is further supported by the lack of obvious endometrial phenotype in the MISR2 KO female mice (Mishina et al., 1996).

3) The authors are quite liberal in using the term progenitor or the multipotent progenitor (subsection “MIS blocks the expansion and differentiation of an Misr2+ stromal progenitor”) cells across the manuscript. However, definitive evidence showing that these MISr2-positive cells are progenitor cell population is not provided. To use term stem/progenitor, the author should temporary label these cells and then track their fate to show these labeled cells give rise to other cell types of the uterus (multipotent) or stromal cells (unipotent progenitors). The experiment with Misr2-cre-tdtomato mice somewhat addresses this issue. However, MISr2 expression is present in the MD stroma from 13.5dpc in mice, and it is not clear whether these cells were from 13.5dpc, when Cre was activated, or day 2-6 postnatal uteri. This could be easily addressed by developing MISr2cre;rttaflox;teo-tdtomato mice where cells can be labeled postnatally and tracked over time. Alternatively, the authors can downplay the emphasis on stem/progenitor cells specially multipotency of these cells.

We agree with this assessment of the reviewer. While our lineage tracing experiment, RNA in situ developmental time series of *Misr2* expression, single cell RNA sequencing atlas, and MIS treatment outcomes strongly support the hypothesis of a direct lineage between the *Misr2*+ subluminal mesenchyme present at birth and the endometrial stroma which develops at PND6, only an inducible MISR2 reporter mouse could provide incontrovertible evidence of this lineage. Thus we have started working on the generation of a Misr2-cre-ert2transgenic mouse line. Given the timeline of transgenic mouse development, and it is therefore beyond the scope of the current manuscript. To ensure we do not overstate our findings we have replaced the usage of “stem/progenitor” with “putative progenitor” throughout the text, and clarified the presumptive nature of the progenitor function (Results paragraph three).

4) The authors need to explain discrepancies in their in-situ data of the MISr2 gene and MISr2-lacZ mice data where real-time lacZ expression is maintained till adulthood in females (Arango et al., 2008)

We have further elaborated on the spatiotemporal pattern of MISR2 expression during Mullerian development in the Results (Figure 1A, Figure 1—figure supplement 1), and reconciled our findings with those of Arango et al. (described in paragraph one of the Results section) by including an additional timepoint of RNA in situ at later time points (See Figure 1A (PND20 in mice) and Figure 1—figure supplement 1E (PND15 in rats). While *Misr2* is expressed in the subluminal mesenchyme embryonically and perinatally, the expression can later be found in the myometrium beyond PND6. Our findings support the hypothesis that the myometrium is not derived from the *Misr2*+ *postnatal*subluminal mesenchyme, but rather precedes the development of the endometrial stroma, and only later acquires expression of *Misr2*.

5) There is no information on how many tissue sections per mice were examined and their relative location (close to the ovary or cervix etc.). How many were used for in-situ? What depth? There are significant differences in histology and the lumen width across the length of uterus.

Numbers of tissue sections per mice per time point analyzed for in situ are now presented in Figure 1—source data 1 and are also summarized in the figure legends of Figure 1A and Figure 1—figure supplement 1E, including additional tissue sections which we recently analyzed (n>12 both for mice and rats). A new figure was also added to demonstrate the relative locations in the uterine horns which were analyzed in the paper (Figure 1—figure supplement 1). In all other tissue sections presented in the paper the middle regions of the horns are used (i.e. “b” in Figure 1—figure supplement 1D), which is now described in the Materials and methods section (subsection “Histomorphological analysis and quantification of RNA-scope images”). Additionally, further *Misr2* in situ analyses were added as a new figure (Figure 1—figure supplement 1D), which confirmed a similar subluminal *Misr2* expression pattern from the fallopian side and the cervix side of the developing uteri.